# DOES ZERO-SHOT REINFORCEMENT LEARNING EXIST?

**Ahmed Touati, Jérémy Rapin & Yann Ollivier**
Meta AI Research, Paris, {atouati,jrapin,yol}@meta.com

## ABSTRACT

A *zero-shot RL agent* is an agent that can solve *any* RL task in a given environment, instantly with no additional planning or learning, after an initial reward-free learning phase. This marks a shift from the reward-centric RL paradigm towards "controllable" agents that can follow arbitrary instructions in an environment. Current RL agents can solve families of related tasks at best, or require planning anew for each task. Strategies for approximate zero-shot RL have been suggested using successor features (SFs) (Borsa et al., 2018) or forward-backward (FB) representations (Touati & Ollivier, 2021), but testing has been limited.

After clarifying the relationships between these schemes, we introduce improved losses and new SF models, and test the viability of zero-shot RL schemes systematically on tasks from the Unsupervised RL benchmark (Laskin et al., 2021). To disentangle universal representation learning from exploration, we work in an offline setting and repeat the tests on several existing replay buffers.

SFs appear to suffer from the choice of the elementary state features. SFs with Laplacian eigenfunctions do well, while SFs based on auto-encoders, inverse curiosity, transition models, low-rank transition matrix, contrastive learning, or diversity (APS), perform unconsistently. In contrast, FB representations jointly learn the elementary and successor features from a single, principled criterion. They perform best and consistently across the board, reaching $85\%$ of supervised RL performance with a good replay buffer, in a zero-shot manner.

## 1 INTRODUCTION

For breadth of applications, reinforcement learning (RL) lags behind other fields of machine learning, such as vision or natural language processing, which have effectively adapted to a wide range of tasks, often in almost zero-shot manner, using pretraining on large, unlabelled datasets (Brown et al., 2020). The RL paradigm itself may be in part to blame: RL agents are usually trained for only one reward function or a small family of related rewards. Instead, we would like to train "controllable" agents that can be given a description of any task (reward function) in their environment, and then immediately know what to do, reacting instantly to such commands as "fetch this object while avoiding that area".

The promise of *zero-shot RL* is to train without rewards or tasks, yet immediately perform well on any reward function given at test time, with no extra training, planning, or finetuning, and only a minimal amount of extra computation to process a task description (Section 2 gives the precise definition we use for *zero-shot RL*). How far away are such zero-shot agents? In the RL paradigm, a new task (reward function) means re-training the agent from scratch, and providing many reward samples. Model-based RL trains a reward-free, task-independent world model, but still requires heavy planning when a new reward function is specified (e.g, Chua et al., 2018; Moerland et al., 2020). Model-free RL is reward-centric from start, and produces specialized agents. Multi-task agents generalize within a family of related tasks only. Reward-free, unsupervised skill pre-training (e.g, Eysenbach et al., 2018) still requires substantial downstream task adaptation, such as training a hierarchical controller.

Is zero-shot RL possible? If one ignores practicality, zero-shot RL is easy: make a list of all possible rewards up to precision $\varepsilon$, then pre-learn all the associated optimal policies. Scalable zero-shot RL must somehow exploit the relationships between policies for all tasks. Learning to go from $a$ to $c$ is not independent from going from $a$ to $b$ and $b$ to $c$, and this produces rich, exploitable algebraic relationships (Blier et al., 2021; Schaul et al., 2015).

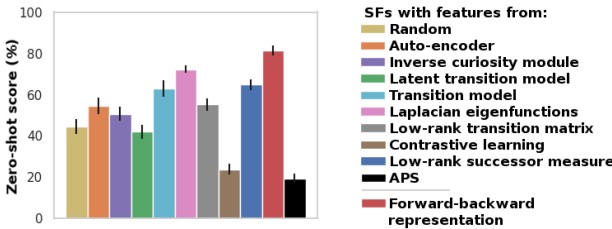

Figure 1: Zero-shot scores of ten SF methods and FB, as a percentage of the supervised score of offline TD3 trained on the same replay buffer, averaged on some tasks, environments and replay buffers from the Unsupervised RL and ExORL benchmarks (Laskin et al., 2022; Yarats et al., 2021). FB and SFs with Laplacian eigenfunctions achieve zero-shot scores approaching supervised RL.

Suggested strategies for generic zero-shot RL so far have used successor representations (Dayan, 1993), under two forms: *successor features* (SFs) (Barreto et al., 2017) as in (Borsa et al., 2018; Hansen et al., 2019; Liu & Abbeel, 2021); and *forward-backward* (FB) representations (Touati & Ollivier, 2021). Both SFs and FB lie in between model-free and model-based RL, by predicting features of future states, or summarizing long-term state-state relationships. Like model-based approaches, they decouple the dynamics of the environment from the reward function. Contrary to world models, they require neither planning at test time nor a generative model of states or trajectories.

Yet SFs heavily depend on a choice of basic state features. To get a full zero-shot RL algorithm, a representation learning method must provide those. While SFs have been successively applied to transfer between tasks, most of the time, the basic features were handcrafted or learned using prior task class knowledge. Meanwhile, FB is a standalone method with no task prior and good theoretical backing, but testing has been limited to goal-reaching in a few environments. Here:

- We systematically assess SFs and FB for zero-shot RL, including many new models of SF basic features, and improved FB loss functions. We use 13 tasks from the Unsupervised RL benchmark (Laskin et al., 2021), repeated on several ExORL training replay buffers (Yarats et al., 2021) to assess robustness to the exploration method.
- We systematically study the influence of basic features for SFs, by testing SFs on features from ten RL representation learning methods. such as latent next state prediction, inverse curiosity module, contrastive learning, diversity, various spectral decompositions...
- We expose new mathematical links between SFs, FB, and other representations in RL.
- We discuss the implicit assumptions and limitations behind zero-shot RL approaches.

## 2 PROBLEM AND NOTATION; DEFINING ZERO-SHOT RL

Let $\mathcal{M} = (S, A, P, \gamma)$ be a reward-free Markov decision process (MDP) with state space $S$, action space $A$, transition probabilities $P(s'|s, a)$ from state $s$ to $s'$ given action $a$, and discount factor $0 < \gamma < 1$ (Sutton & Barto, 2018). If $S$ and $A$ are finite, $P(s'|s, a)$ can be viewed as a stochastic matrix $P_{sas'} \in \mathbb{R}^{(|S| \times |A|) \times |S|}$; in general, for each $(s, a) \in S \times A$, $P(\mathrm{d}s'|s, a)$ is a probability measure on $s' \in S$. The notation $P(\mathrm{d}s'|s, a)$ covers all cases. Given $(s_0, a_0) \in S \times A$ and a policy $\pi \colon S \to \mathrm{Prob}(A)$, we denote $\mathrm{Pr}(\cdot|s_0, a_0, \pi)$ and $\mathbb{E}[\cdot|s_0, a_0, \pi]$ the probabilities and expectations under state-action sequences $(s_t, a_t)_{t \geq 0}$ starting at $(s_0, a_0)$ and following policy $\pi$ in the environment, defined by sampling $s_t \sim P(\mathrm{d}s_t|s_{t-1}, a_{t-1})$ and $a_t \sim \pi(\mathrm{d}a_t|s_t)$. We define $P_\pi(\mathrm{d}s', \mathrm{d}a'|s, a) := P(\mathrm{d}s'|s, a)\pi(\mathrm{d}a'|s')$ and $P_\pi(\mathrm{d}s'|s) := \int P(\mathrm{d}s'|s, a)\pi(\mathrm{d}a|s)$, the state-action transition probabilities and state transition probabilities induced by $\pi$. Given a reward function $r \colon S \to \mathbb{R}$, the $Q$-function of $\pi$ for $r$ is $Q_r^\pi(s_0, a_0) := \sum_{t \geq 0} \gamma^t \mathbb{E}[r(s_{t+1})|s_0, a_0, \pi]$. For simplicity, we assume the reward $r$ depends only on the next state $s_{t+1}$ instead on the full triplet $(s_t, a_t, s_{t+1})$, but this is not essential.

We focus on offline unsupervised RL, where the agent cannot interact with the environment. The agent only has access to a static dataset of logged reward-free transitions in the environment, $\mathcal{D} = \{(s_i, a_i, s_i')\}_{i \in \mathcal{I}}$ with $s_i' \sim P(\mathrm{d}s_i'|s_i, a_i)$. These can come from any exploration method or methods.

The offline setting disentangles the effects of the exploration method and representation and policy learning: we test each zero-shot method on several training datasets from several exploration methods.

We denote by $\rho(\mathrm{d}s)$ and $\rho(\mathrm{d}s, \mathrm{d}a)$ the (unknown) marginal distribution of states and state-actions in the dataset $\mathcal{D}$. We use both $\mathbb{E}_{s \sim \mathcal{D}}[\cdot]$ and $\mathbb{E}_{s \sim \rho}[\cdot]$ for expectations under the training distribution.

**Zero-shot RL: problem statement.** The goal of zero-shot RL is to compute a compact representation $\mathcal{E}$ of the environment by observing samples of reward-free transitions $(s_t, a_t, s_{t+1})$ in this environment. Once a reward function is specified later, the agent must use $\mathcal{E}$ to immediately produce a good policy, via only elementary computations without any further planning or learning. Ideally, for any downstream task, the performance of the returned policy should be close to the performance of a supervised RL baseline trained on the same dataset labeled with the rewards for that task.

Reward functions may be specified at test time either as a relatively small set of reward samples $(s_i, r_i)$, or as an explicit function $s \mapsto r(s)$ (such as $1$ at a known goal state and $0$ elsewhere). The method will be few-shot, zero-planning in the first case, and truly zero-shot in the second case.

## 3 RELATED WORK

Zero-shot RL requires unsupervised learning and the absence of any planning or fine-tuning at test time. The proposed strategies for zero-shot RL discussed in Section 1 ultimately derive from successor representations (Dayan, 1993) in finite spaces. In continuous spaces, starting with a finite number of features $\varphi$, successor features can be used to produce policies within a family of tasks directly related to $\varphi$ (Barreto et al., 2017; Borsa et al., 2018; Zhang et al., 2017; Grimm et al., 2019), often using hand-crafted $\varphi$ or learning $\varphi$ that best linearize training rewards. VISR (Hansen et al., 2019) and its successor APS (Liu & Abbeel, 2021) use SFs with $\varphi$ automatically built online via diversity criteria (Eysenbach et al., 2018; Gregor et al., 2016). We include APS among our baselines, as well as many new criteria to build $\varphi$ automatically.

Successor measures (Blier et al., 2021) avoid the need for $\varphi$ by directly learning models of the distribution of future states: doing this for various policies yields a candidate zero-shot RL method, forward-backward representations (Touati & Ollivier, 2021), which has been tested for goal-reaching in a few environments with discrete actions. FB uses a low-rank model of long-term state-state relationships reminiscent of the state-goal factorization from Schaul et al. (2015).

Model-based RL (surveyed in Moerland et al. (2020)) misses the zero-planning requirement of zero-shot RL. Still, learned models of the transitions between states can be used jointly with SFs to provide zero-shot methods (**Trans**, **Latent**, and **LRA-P** methods below).

Goal-oriented and multitask RL has a long history (e.g, Foster & Dayan, 2002; Sutton et al., 2011; da Silva et al., 2012; Schaul et al., 2015; Andrychowicz et al., 2017). A parametric family of tasks must be defined in advance (e.g., reaching arbitrary goal states). New rewards cannot be set a posteriori: for example, a goal-state-oriented method cannot handle dense rewards. Zero-shot task transfer methods learn on tasks and can transfer to related tasks only (e.g, Oh et al., 2017; Sohn et al., 2018); this can be used, e.g., for sim-to-real transfer (Genc et al., 2020) or slight environment changes, which is not covered here. Instead, we aim at not having any predefined family of tasks.

Unsupervised skill and option discovery methods, based for instance on diversity (Eysenbach et al., 2018; Gregor et al., 2016) or eigenoptions (Machado et al., 2017) can learn a variety of behaviors without rewards. Downstream tasks require learning a hierarchical controller to combine the right skills or options for each task. Directly using unmodified skills has limited performance without heavy finetuning (Eysenbach et al., 2018). Still, these methods can speed up downstream learning.

The unsupervised aspect of some of these methods (including DIAYN and APS) has been disputed, because training still used end-of-trajectory signals, which are directly correlated to the downstream task in some common environments: without this signal, results drop sharply (Laskin et al., 2022).

## 4 SUCCESSOR REPRESENTATIONS AND ZERO-SHOT RL

For a finite MDP, the *successor representation* (SR) (Dayan, 1993) $M^\pi(s_0, a_0)$ of a state-action pair $(s_0, a_0)$ under a policy $\pi$, is defined as the discounted sum of future occurrences of each state:

$$M^\pi(s_0, a_0, s) := \mathbb{E}\left[\sum_{t \geq 0} \gamma^t \mathbb{1}_{\{s_{t+1} = s\}} \mid (s_0, a_0), \pi\right] \quad \forall s \in S. \tag{1}$$

In matrix form, SRs can be written as $M^\pi = P \sum_{t \geq 0} \gamma^t P_\pi^t = P(\mathrm{Id} - \gamma P_\pi)^{-1}$, where $P_\pi$ is the state transition probability. $M^\pi$ satisfies the matrix Bellman equation $M^\pi = P + \gamma P_\pi M^\pi$.

Importantly, SRs disentangle the dynamics of the MDP and the reward function: for any reward $r$ and policy $\pi$, the $Q$-function can be expressed linearly as $Q_r^\pi = M^\pi r$.

**Successor features and successor measures.** *Successor features* (SFs) (Barreto et al., 2017) extend SR to continous MDPs by first assuming we are given a *basic feature* map $\varphi \colon S \to \mathbb{R}^d$ that embeds states into $d$-dimensional space, and defining the expected discounted sum of future state features:

$$\psi^\pi(s_0, a_0) := \mathbb{E}\left[\sum_{t \geq 0} \gamma^t \varphi(s_{t+1}) \mid s_0, a_0, \pi\right]. \tag{2}$$

SFs have been introduced to make SRs compatible with function approximation. For a finite MDP, the original definition (1) is recovered by letting $\varphi$ be a one-hot state encoding into $\mathbb{R}^{|S|}$.

Alternatively, *successor measures* (SMs) (Blier et al., 2021) extend SRs to continuous spaces by treating the distribution of future visited states as a measure $M^\pi$ over the state space $S$,

$$M^\pi(s_0, a_0, X) := \sum_{t \geq 0} \gamma^t \Pr\left(s_{t+1} \in X \mid s_0, a_0, \pi\right) \quad \forall X \subset S. \tag{3}$$

SFs and SMs are related: by construction, $\psi^\pi(s_0, a_0) = \int_{s'} M^\pi(s_0, a_0, \mathrm{d}s')\, \varphi(s')$.

**Zero-shot RL from successor features and forward-backward representations.** Successor representations provide a generic framework for zero-shot RL, by learning to represent the relationship between reward functions and $Q$-functions, as encoded in $M^\pi$.

Given a basic feature map $\varphi \colon S \to \mathbb{R}^d$ to be learned via another criterion, *universal SFs* (Borsa et al., 2018) learn the successor features of a particular family of policies $\pi_z$ for $z \in \mathbb{R}^d$,

$$\psi(s_0, a_0, z) = \mathbb{E}\left[\sum_{t \geq 0} \gamma^t \varphi(s_{t+1}) \mid (s_0, a_0), \pi_z\right], \quad \pi_z(s) := \arg\max_a \psi(s, a, z)^\top z. \tag{4}$$

Once a reward function $r$ is revealed, we use a few reward samples or explicit knowledge of the function $r$ to perform a linear regression of $r$ onto the features $\varphi$. Namely, we estimate $z_r := \arg\min_z \mathbb{E}_{s \sim \rho}[(r(s) - \varphi(s)^\top z)^2] = \mathbb{E}_\rho[\varphi\varphi^\top]^{-1} \mathbb{E}_\rho[\varphi r]$. Then we return the policy $\pi_{z_r}$. This policy is guaranteed to be optimal for all rewards in the linear span of the features $\varphi$:

**Theorem 1** (Borsa et al. (2018)). *Assume that* (4) *holds. Assume there exists a weight* $w \in \mathbb{R}^d$ *such that* $r(s) = \varphi(s)^\top w, \forall s \in S$. *Then* $z_r = w$, *and* $\pi_{z_r}$ *is the optimal policy for reward* $r$.

*Forward-backward (FB) representations* (Touati & Ollivier, 2021) apply a similar idea to a finite-rank model of successor measures. They look for representations $F \colon S \times A \times \mathbb{R}^d \to \mathbb{R}^d$ and $B \colon S \to \mathbb{R}^d$ such that the long-term transition probabilities $M^{\pi_z}$ in (3) decompose as

$$M^{\pi_z}(s_0, a_0, \mathrm{d}s') \approx F(s_0, a_0, z)^\top B(s')\, \rho(\mathrm{d}s'), \quad \pi_z(s) := \arg\max_a F(s, a, z)^\top z \tag{5}$$

In a finite space, the first equation rewrites as the matrix decomposition $M^{\pi_z} = F_z^\top B \operatorname{diag}(\rho)$.

Once a reward function $r$ is revealed, we estimate $z_r := \mathbb{E}_{s \sim \rho}[r(s)B(s)]$ from a few reward samples or from explicit knowledge of the function $r$ (e.g. $z_r = B(s)$ to reach $s$). Then we return the policy $\pi_{z_r}$. If the approximation (5) holds, this policy is guaranteed to be optimal for any reward function:

**Theorem 2** (Touati & Ollivier (2021)). *Assume that* (5) *holds. Then for any reward function* $r$, *the policy* $\pi_{z_r}$ *is optimal for* $r$, *with optimal $Q$-function* $Q_r^\star = F(s, a, z_r)^\top z_r$.

For completeness, we sketch the proofs of Theorems 1–2 in Appendix A. Importantly, both theorems are compatible with approximation: approximate solutions provide approximately optimal policies.

**Connections between SFs and FB.** A first difference between SFs and FB is that SFs must be provided with basic features $\varphi$. The best $\varphi$ is such that the reward functions of the downstream tasks are linear in $\varphi$. But for unsupervised training without prior task knowledge, an external criterion is needed to learn $\varphi$. We test a series of such criteria below. In contrast, FB uses a single criterion, avoiding the need for state featurization by learning a model of state occupancy.

Second, SFs only cover rewards in the linear span of $\varphi$, while FB apparently covers any reward. But this difference is not as stark as it looks: exactly solving the FB equation (5) in continuous spaces

requires $d = \infty$, and for finite $d$, the policies will only be optimal for rewards in the linear span of $B$ (Touati & Ollivier, 2021). Thus, in both cases, policies are exactly optimal only for a $d$-dimensional family of rewards. Still, FB can use an arbitrary large $d$ without any additional input or criterion.

FB representations are related to successor features: the FB definition (5) implies that $\psi(s, a, z) := F(s, a, z)$ are the successor features of $\varphi(s) := (\mathbb{E}_\rho BB^\top)^{-1}B(s)$. This follows from multiplying (3) and (5) by $B^\top(\mathbb{E}_\rho BB^\top)^{-1}$ on the right, and integrating over $s' \sim \rho$. Thus, a posteriori, FB can be used to produce both $\varphi$ and $\psi$ in SF, although training is different.

This connection between FB and SF is one-directional: (5) is a stronger condition. In particular $F = B = 0$ is not a solution: contrary to $\psi = \varphi = 0$ in (4), there is no collapse. No additional criterion to train $\varphi$ is required: $F$ and $B$ are trained jointly to provide the best rank-$d$ approximation of the successor measures $M^\pi$. This summarizes an environment by selecting the features that best describe the relationship $Q_r^\pi = M^\pi r$ between rewards and $Q$-functions.

# 5 ALGORITHMS FOR SUCCESSOR FEATURES AND FB REPRESENTATIONS

We now describe more precisely the algorithms used in our experiments. The losses used to train $\psi$ in SFs, and $F, B$ in FB, are described in Sections 5.1 and 5.2 respectively.

To obtain a full zero-shot RL algorithm, SFs must specify the basic features $\varphi$. Any representation learning method can be used for $\varphi$. We use ten possible choices (Section 5.3) based on existing or new representations for RL: random features as a baseline, autoencoders, next state and latent next state transition models, inverse curiosity module, the diversity criterion of APS, contrastive learning, and finally, several spectral decompositions of the transition matrix or its associated Laplacian.

Both SFs and FB define policies as an argmax of $\psi(s, a, z)^\top z$ or $F(s, a, z)^\top z$ over actions $a$. With continuous actions, the argmax cannot be computed exactly. We train an auxiliary policy network $\pi(s, z)$ to approximate this argmax, using the same standard method for SFs and FB (Appendix G.4).

## 5.1 LEARNING THE SUCCESSOR FEATURES $\psi$

The successor features $\psi$ satisfy the $\mathbb{R}^d$-valued Bellman equation $\psi^\pi = P\varphi + \gamma P_\pi \psi^\pi$, the collection of ordinary Bellman equations for each component of $\varphi$. The $P$ in front of $\varphi$ comes from using $\varphi(s_{t+1})$ not $\varphi(s_t)$ in (2). Therefore, we can train $\psi(s, a, z)$ for each $z$ by minimizing the Bellman residuals $\left\|\psi(s_t, a_t, z) - \varphi(s_{t+1}) - \gamma\bar{\psi}(s_{t+1}, \pi_z(s_{t+1}), z)\right\|^2$ where $\bar\psi$ is a non-trainable target version of $\psi$ as in parametric $Q$-learning. This requires sampling a transition $(s_t, a_t, s_{t+1})$ from the dataset and choosing $z$. We sample random values of $z$ as described in Appendix G.3.

This is the loss used in Borsa et al. (2018). But this can be improved, since we do not use the full vector $\psi(s, a, z)$: only $\psi(s, a, z)^\top z$ is needed for the policies. Therefore, as in Liu & Abbeel (2021), instead of the vector-valued Bellman residual above, we just use

$$\mathcal{L}(\psi) := \mathbb{E}_{(s_t, a_t, s_{t+1})\sim\rho}\left(\psi(s_t, a_t, z)^\top z - \varphi(s_{t+1})^\top z - \gamma\bar\psi(s_{t+1}, \pi_z(s_{t+1}), z)^\top z\right)^2 \quad (6)$$

for each $z$. This trains $\psi(\cdot, z)^\top z$ as the $Q$-function of reward $\varphi^\top z$, the only case needed, while training the full vector $\psi(\cdot, z)$ amounts to training the $Q$-functions of each policy $\pi_z$ for all rewards $\varphi^\top z'$ for all $z' \in \mathbb{R}^d$ including $z' \neq z$. We have found this improves performance.

## 5.2 LEARNING FB REPRESENTATIONS: THE FB TRAINING LOSS

The successor measure $M^\pi$ satisfies a Bellman-like equation $M^\pi = P + \gamma P_\pi M^\pi$, as matrices in the finite case and as measures in the general case (Blier et al., 2021). We can learn FB by iteratively minimizing the Bellman residual on the parametric model $M = F^\top B\rho$. Using a suitable norm $\|\cdot\|_\rho$ for the Bellman residual (Appendix B) leads to a loss expressed as expectations from the dataset:

$$\mathcal{L}(F, B) := \left\|F_z^\top B\rho - \left(P + \gamma P_{\pi_z}\bar{F}_z^\top\bar{B}\rho\right)\right\|_\rho^2 \quad (7)$$

$$= \mathbb{E}_{\substack{(s_t, a_t, s_{t+1})\sim\rho \\ s'\sim\rho}}\left[\left(F(s_t, a_t, z)^\top B(s') - \gamma\bar{F}(s_{t+1}, \pi_z(s_{t+1}), z)^\top\bar{B}(s')\right)^2\right]$$

$$- 2\,\mathbb{E}_{(s_t, a_t, s_{t+1})\sim\rho}\left[F(s_t, a_t, z)^\top B(s_{t+1})\right] + \texttt{Const} \quad (8)$$

where the constant term does not depend on $F$ and $B$, and where as usual $\bar{F}_z$ and $\bar{B}$ are non-trainable target versions of $F$ and $B$ whose parameters are updated with a slow-moving average of those of $F$ and $B$. Appendix B quickly derives this loss, with pseudocode in Appendix L. Contrary to Touati & Ollivier (2021), the last term involves $B(s_{t+1})$ instead of $B(s_t)$, because we use $s_{t+1}$ instead of $s_t$ for the successor measures (3). We sample random values of $z$ as described in Appendix G.3.

We include an auxiliary loss (Appendix B) to normalize the covariance of $B$, $\mathbb{E}_\rho BB^\top \approx \mathrm{Id}$, as in Touati & Ollivier (2021) (otherwise one can, e.g., scale $F$ up and $B$ down since only $F^\top B$ is fixed).

As with SFs above, only $F(\cdot, z)^\top z$ is needed for the policies, while the loss above on the vector $F$ amounts to training $F(\cdot, z)^\top z'$ for all pairs $(z, z')$. The full loss is needed for joint training of $F$ and $B$. But we include an auxiliary loss $\mathcal{L}'(F)$ to focus training on the diagonal $z' = z$. This is obtained by multiplying the Bellman gap in $\mathcal{L}$ by $B^\top (B\rho B^\top)^{-1} z$ on the right, to make $F(\cdot, z)^\top z$ appear:

$$\mathcal{L}'(F) := \mathbb{E}_{(s_t, a_t, s_{t+1}) \sim \rho} \left[ \left( F(s_t, a_t, z)^\top z - B(s_{t+1})^\top (\mathbb{E}_\rho BB^\top)^{-1} z - \gamma \bar{F}(s_{t+1}, \pi_z(s_{t+1}), z)^\top z \right)^2 \right].$$
(9)

This trains $F(\cdot, z)^\top z$ as the $Q$-function for reward $B^\top (\mathbb{E}_\rho BB^\top)^{-1} z$. Though $\mathcal{L} = 0$ implies $\mathcal{L}' = 0$, adding $\mathcal{L}'$ reduces the error on the part used for policies. This departs from Touati & Ollivier (2021).

### 5.3  Learning Basic Features $\varphi$ for Successor Features

SFs must be provided with basic state features $\varphi$. Any representation learning method can be used to supply $\varphi$. We focus on prominent RL representation learning baselines, and on those used in previous zero-shot RL candidates such as APS. We now describe the precise learning objective for each.

**Random Features (`Rand`).** We use a non-trainable randomly initialized network as features.

**Autoencoder (`AEnc`).** We learn a decoder $f \colon \mathbb{R}^d \to S$ to recover the state from its representation $\varphi$:

$$\min_{f, \varphi} \mathbb{E}_{s \sim \mathcal{D}}[(f(\varphi(s)) - s)^2].$$
(10)

**Inverse Curiosity Module (`ICM`)** aims at extracting the controllable aspects of the environment (Pathak et al., 2017). The idea is to train an inverse dynamics model $g \colon \mathbb{R}^d \times \mathbb{R}^d \to A$ to predict the action used for a transition between two consecutive states. We use the loss

$$\min_{g, \varphi} \mathbb{E}_{(s_t, a_t, s_{t+1}) \sim \mathcal{D}}[\|g(\varphi(s_t), \varphi(s_{t+1})) - a_t\|^2].$$
(11)

**Transition model (`Trans`).** This is a one-step forward dynamic model $f \colon \mathbb{R}^d \times A \to S$ that predicts the next state from the current state representation:

$$\min_{f, \varphi} \mathbb{E}_{(s_t, a_t, s_{t+1}) \sim \mathcal{D}}[(f(\varphi(s_t), a_t) - s_{t+1})^2].$$
(12)

**Latent transition model (`Latent`).** This is similar to the transition model but instead of predicting the next state, it predicts its representation:

$$\min_{f, \varphi} \mathbb{E}_{(s_t, a_t, s_{t+1}) \sim \mathcal{D}}[(f(\varphi(s_t), a_t) - \varphi(s_{t+1}))^2].$$
(13)

A clear failure case of this loss is when all states are mapped to the same representation. To avoid this collapse, we compute $\varphi(s_{t+1})$ using a non-trainable version of $\varphi$, with parameters corresponding to a slowly moving average of the parameters of $\varphi$, similarly to BYOL (Grill et al., 2020).

**Diversity methods (`APS`).** VISR (Hansen et al., 2019) and its successor APS (Liu & Abbeel, 2021) tackle zero-shot RL using SFs with features $\varphi$ built online from a diversity criterion. This criterion maximizes the mutual information between a policy parameter and the features of the states visited by a policy using that parameter (Eysenbach et al., 2018; Gregor et al., 2016). VISR and APS use, respectively, a variational or nearest-neighbor estimator for the mutual information. We directly use the code provided for APS, and refer to (Liu & Abbeel, 2021) for the details. Contrary to other methods, APS is not offline: it needs to be trained on its own replay buffer.

**Laplacian Eigenfunctions (`Lap`).** Wu et al. (2018) consider the symmetrized MDP graph Laplacian induced by an exploratory policy $\pi$, defined as $\mathcal{L} = \mathrm{Id} - \frac{1}{2}(P_\pi \mathrm{diag}(\rho)^{-1} + \mathrm{diag}(\rho)^{-1}(P_\pi)^\top)$. They

propose to learn the eigenfunctions of $\mathcal{L}$ via the spectral graph drawing objective (Koren, 2003):

$$\min_\varphi \mathbb{E}_{(s_t,s_{t+1})\sim\mathcal{D}}\left[\|\varphi(s_t)-\varphi(s_{t+1})\|^2\right] + \lambda\,\mathbb{E}_{\substack{s\sim\mathcal{D}\\s'\sim\mathcal{D}}}\left[(\varphi(s)^\top\varphi(s'))^2 - \|\varphi(s)\|_2^2 - \|\varphi(s')\|_2^2\right] \quad (14)$$

where the second term is an orthonormality regularization to ensure that $\mathbb{E}_{s\sim\rho}[\varphi(s)\varphi(s)^\top] \approx \mathrm{Id}$, and $\lambda > 0$ is the regularization weight. This is implicitly contrastive, pushing features of $s_t$ and $s_{t+1}$ closer while keeping features apart overall. Such eigenfunctions have long been argued to play a key role in RL (Mahadevan & Maggioni, 2007; Machado et al., 2017).

**Low-Rank Approximation of $P$ (`LRA-P`):** we learn features by estimating a low-rank model of the transition probability densities: $P(\mathrm{d}s'|s,a) \approx \chi(s,a)^\top\mu(s')\,\rho(\mathrm{d}s')$. Knowing $\rho$ is not needed: the corresponding loss on $\chi^\top\mu - P/\rho$ is readily expressed as expectations over the dataset,

$$\min_{\chi,\mu} \mathbb{E}_{\substack{(s_t,a_t)\sim\rho\\s'\sim\rho}}\left[\left(\chi(s_t,a_t)^\top\mu(s') - \frac{P(\mathrm{d}s'|s_t,a_t)}{\rho(\mathrm{d}s')}\right)^2\right] \quad (15)$$

$$= \mathbb{E}_{\substack{(s_t,a_t)\sim\rho\\s'\sim\rho}}[(\chi(s_t,a_t)^\top\mu(s'))^2] - 2\,\mathbb{E}_{(s_t,a_t,s_{t+1})\sim\rho}[\chi(s_t,a_t)^\top\mu(s_{t+1})] + \texttt{Const} \quad (16)$$

We normalize $\mathbb{E}_\rho[\mu\mu^\top] \approx \mathrm{Id}$ with the same loss used for $B$. Then we use SFs with $\varphi := \mu$. If the model $P = \chi^\top\mu\,\rho$ is exact, this provides exact optimal policies for any reward (Appendix E, Thm. 3).

This loss is implicitly contrastive: it compares samples $s_{t+1}$ to independent samples $s'$ from $\rho$. It is an asymmetric extension of the Laplacian loss (14) (Appendix F). The loss (16) is also a special case of the FB loss (8) by setting $\gamma = 0$, omitting $z$, and substituting $(\chi,\mu)$ for $(F, B)$. Indeed, FB learns a finite-rank model of $P(\mathrm{Id}-\gamma P_\pi)^{-1}$, which equals $P$ when $\gamma = 0$.

A related loss is introduced in Ren et al. (2022), but involves a second unspecified, arbitrary probability distribution $p$, which must cover the whole state space and whose analytic expression must be known. It is unclear how to set a suitable $p$ in general.

**Contrastive Learning (`CL`)** methods learn representations by pushing positive pairs (similar states) closer together while keeping negative pairs apart. Here, two states are considered similar if they lie close on the same trajectory. We use a SimCLR-like objective (Chen et al., 2020):

$$\min_{\chi,\varphi} - \mathbb{E}_{\substack{k\sim\mathrm{Geom}(1-\gamma_{\texttt{CL}})\\(s_t,s_{t+k})\sim\mathcal{D}}}\left[\log\frac{\exp(\texttt{cosine}(\chi(s_t),\varphi(s_{t+k})))}{\mathbb{E}_{s'\sim\mathcal{D}}\exp(\texttt{cosine}(\chi(s_t),\varphi(s')))}\right] \quad (17)$$

where $s_{t+k}$ is the state encountered at step $t+k$ along the subtrajectory that starts at $s_t$, where $k$ is sampled from a geometric distribution of parameter $(1 - \gamma_{\texttt{CL}})$, and $\texttt{cosine}(u,v) = \frac{u^\top v}{\|u\|_2\|v\|_2}, \forall u, v \in \mathbb{R}^d$ is the cosine similarity function. Here $\gamma_{\texttt{CL}} \in [0; 1)$ is a parameter not necessarily set to the MDP's discount factor $\gamma$. `CL` requires a dataset made of full trajectories instead of isolated transitions.

`CL` is tightly related to the spectral decomposition of the successor measure $\sum_t \gamma_{\texttt{CL}}^t P_\pi^{t+1}$, where $\pi$ is the behavior policy generating the dataset trajectories. Precisely, assuming that $\chi$ and $\varphi$ are centered with unit norm, and expanding the $\log$ and $\exp$ at second order, the loss (17) becomes

$$(17) \approx \tfrac{1}{2}\,\mathbb{E}_{\substack{s\sim\mathcal{D}\\s'\sim\mathcal{D}}}[(\chi(s)^\top\varphi(s'))^2] - \mathbb{E}_{\substack{k\sim\mathrm{Geom}(1-\gamma_{\texttt{CL}})\\(s_t,s_{t+k})\sim\mathcal{D}}}\left[\chi(s_t)^\top\varphi(s_{t+k})\right] \quad (18)$$

(compare (16)). Now, the law of $s_{t+k}$ given $s_t$ is given by the stochastic matrix $(1-\gamma_{\texttt{CL}})\sum_t \gamma_{\texttt{CL}}^t P_\pi^{t+1}$, the rescaled successor measure of $\pi$. Then one finds that (18) is minimized when $\chi$ and $\varphi$ provide the singular value decomposition of this matrix in $L^2(\rho)$ norm (Appendix C). Formal links between contrastive learning and spectral methods can be found in Tian (2022); Balestriero & LeCun (2022).

**Low-Rank Approximation of SR (`LRA-SR`).** The `CL` method implicitly factorizes the successor measure of the exploration policy in Monte Carlo fashion by sampling pairs $(s_t, s_{t+k})$ on the same trajectory. This may suffer from high variance. To mitigate this, we propose to factorize this successor measure by temporal difference learning instead of Monte Carlo. This is achieved with an FB-like loss (8) except we drop the policies $\pi_z$ and learn successor measures for the exploration policy only:

$$\min_{\chi,\varphi} \mathbb{E}_{\substack{(s_t,s_{t+1})\sim\mathcal{D}\\s'\sim\mathcal{D}}}\left[\left(\chi(s_t)^\top\varphi(s') - \gamma\bar{\chi}(s_{t+1})^\top\bar{\varphi}(s')\right)^2\right] - 2\,\mathbb{E}_{(s_t,s_{t+1})\sim\mathcal{D}}\left[\chi(s_t)^\top\varphi(s_{t+1}))\right] \quad (19)$$

with $\bar{\chi}$ and $\bar{\varphi}$ target versions of $\chi$ and $\varphi$. We normalize $\varphi$ to $\mathbb{E}_\rho[\varphi\varphi^\top] \approx \mathrm{Id}$ with the same loss as for $B$. Of all SF variants tested, this is the closest to FB.

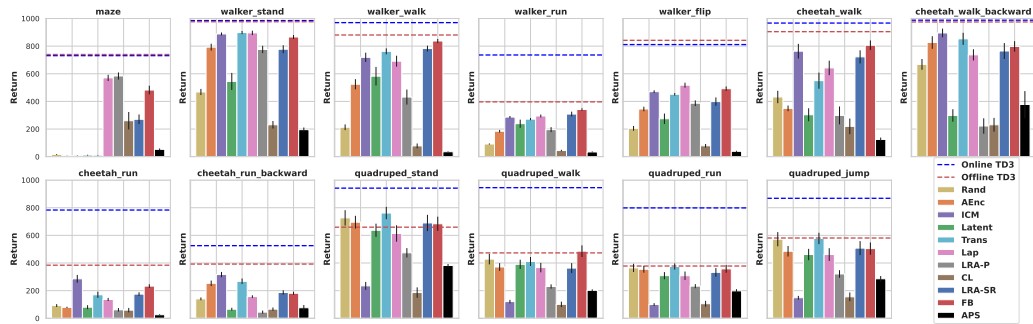

Figure 2: Zero-shot scores for each task, with supervised online and offline TD3 as toplines. Average over 3 replay buffers and 10 random seeds.

## 6 EXPERIMENTAL RESULTS ON BENCHMARKS

Each of the 11 methods (FB and 10 SF-based models) has been tested on 13 tasks in 4 environments from the Unsupervised RL and ExORL benchmarks (Laskin et al., 2021; Yarats et al., 2021): Maze (reach 20 goals), Walker (stand, walk, run, flip), Cheetah (walk, run, walk backwards, run backwards), and Quadruped (stand, walk, run, jump); see Appendix G.1. Each task and method was repeated for 3 choices of replay buffer from ExORL: RND, APS, and Proto (except for the APS method, which can only train on the APS buffer). Each of these 403 settings was repeated with 10 random seeds.

The full setup is described in Appendix G. Representation dimension is $d = 50$, except for Maze ($d = 100$). After model training, tasks are revealed by 10,000 reward samples as in (Liu & Abbeel, 2021), except for Maze, where a known goal is presented and used to set $z_r$ directly. The code can be found at https://github.com/facebookresearch/controllable_agent .

As toplines, we use online TD3 (with task rewards, and free environment interactions not restricted to a replay buffer), and offline TD3 (restricted to each replay buffer labelled with task rewards). Offline TD3 gives an idea of the best achievable performance given the training data in a buffer.

In Fig. 2 we plot the performance of each method for each task in each environment, averaged over the three replay buffers and ten random seeds. Appendix H contains the full results and more plots.

Compared to offline TD3 as a reference, on the Maze tasks, **Lap**, **LRA-P**, and **FB** perform well. On the Walker tasks, **ICM**, **Trans**, **Lap**, **LRA-SR**, and **FB** perform well. On the Cheetah tasks, **ICM**, **Trans**, **Lap**, **LRA-SR**, and **FB** perform well. On the Quadruped tasks, many methods perform well, including, surprisingly, random features. Appendix I plots some of the learned features on Maze.

On Maze, none of the encoder-based losses learn good policies, contrary to FB and spectral SF methods. We believe this is because $(x, y)$ is already a good representation of the 2D state $s$, so the encoders do nothing. Yet SFs on $(x, y)$ cannot solve the task (SFs on rewards of the form $ax + by$ don't recover goal-oriented tasks): planning with SFs requires specific representations.

Fig. 1 reports aggregated scores over all tasks. To average across tasks, we normalize scores: for each task and replay buffer, performance is expressed as a percentage of the performance of offline TD3 on the same replay buffer, a natural supervised topline given the data. These normalized scores are averaged over all tasks in each environment, then over environments to yield the scores in Fig. 1. The variations over environments, replay buffers and random seeds are reported in Appendix H.

These results are broadly consistent over replay buffers. Buffer-specific results are reported in Appendices H.3–H.4. Sometimes a replay buffer is restrictive, as attested by poor offline TD3 performance, starkly so for the Proto buffer on all Quadruped tasks. APS does not work well as a zero-shot RL method, but it does work well as an exploration method: on average, the APS and RND replay buffers have close results.

**FB** and **Lap** are the only methods that perform consistently well, both over tasks and over replay buffers. Averaged on all tasks and buffers, **FB** reaches 81% of supervised offline TD3 performance, and 85% on the RND buffer. The second-best method is **Lap** with 74% (78% on the Proto buffer).

## 7 DISCUSSION AND LIMITATIONS

**Can a few features solve many reward functions? Why are some features better?** The choice of features is critical. Consider goal-reaching tasks: the obvious way to learn to reach arbitrary goal states via SFs is to define one feature per possible goal state (one-hot encoding). This requires $|S|$ features, and does not scale to continuous spaces. But much better features exist. For instance, with an size-$n$ cycle $S = \{0, \ldots, n-1\}$ with actions that move left and right modulo $n$, then *two* features suffice instead of $n$: SFs with $\varphi(s) = (\cos(2\pi s/n), \sin(2\pi s/n))$ provides exact optimal policies to reach any arbitrary state. On a $d$-dimensional grid $S = \{0, \ldots, n-1\}^d$, just $2d$ features (a sine and cosine in each direction) are sufficient to reach any of the $n^d$ goal states via SFs.

Goal-reaching is only a subset of possible RL tasks, but this clearly shows that some features are better than others. The sine and cosine are the main eigenfunctions of the graph Laplacian on the grid: such features have long been argued to play a special role in RL (Mahadevan & Maggioni, 2007; Machado et al., 2017). FB-based methods are theoretically known to learn such eigenfunctions (Blier et al., 2021). Yet a precise theoretical link to downstream performance is still lacking.

**Are these finite-rank models reasonable?** FB crucially relies on a finite-rank model of $\sum \gamma^t P_\pi^t$, while some SF variants above rely on finite-rank models of $P$ or the corresponding Laplacian. It turns out such approximations are very different for $P$ or for $\sum \gamma^t P_\pi^t$.

Unfortunately, despite the popularity of low-rank $P$ assumptions in the theoretical literature, $P_\pi$ is *always* close to Id in situations where $s_{t+1}$ is close to $s_t$, such as any continuous-time physical system (Appendix D). Any low-rank model of $P_\pi$ will be poor; actually $P_\pi$ is better modeled as Id − low-rank, thus approximating the Laplacian. On the other hand, $P_\pi^t$ with large $t$ gets close to rank one under weak assumptions (ergodicity), as $P_\pi^t$ converges to an equilibrium distribution when $t \to \infty$. Thus the spectrum of the successor measures $\sum \gamma^t P_\pi^t$ is usually more spread out (details and examples in Appendix D), and a low-rank model makes sense. The eigenvalues of $P_\pi$ are close to 1 and there is little signal to differentiate between eigenvectors, but differences become clearer over time on $P_\pi^t$. This may explain the better performance of **FB** and **LRA−SR** compared to **LRA−P**.

**Limitations.** First, these experiments are still small-scale and performance is not perfect, so there is space for improvement. Also, all the environments tested here were deterministic: all algorithms still make sense in stochastic environments, but experimental conclusions may differ.

Second, even though these methods can be coupled with any exploration technique, this will obviously not cover tasks too different from the actions in the replay buffer, as with any offline RL method.

These zero-shot RL algorithms learn to summarize the long-term future for a wide range of policies (though without synthesizing trajectories). This is a lot: in contrast, world models only learn policy-independent one-step transitions. So the question remains of how far this can scale. A priori, there is no restriction on the inputs (e.g., images, state history...). Still, for large problems, some form of prior seems unavoidable. For SFs, priors can be integrated in $\varphi$, but rewards must be linear in $\varphi$. For FB, priors can be integrated in $B$'s input. Touati & Ollivier (2021) use FB with pixel-based inputs for $F$, but only the agent's $(x, y)$ position for $B$'s input: this recovers all rewards that are functions of $(x, y)$ (linear or not). Breaking the symmetry of $F$ and $B$ reduces the strain on the model by restricting predictions to fewer variables, such as an agent's future state instead of the full environment.

## 8 CONCLUSIONS

Zero-shot RL methods avoid test-time planning by summarizing long-term state-state relationships for particular policies. We systematically tested forward-backward representations and many new models of successor features on zero-shot tasks. We also uncovered algebraic links between SFs, FB, contrastive learning, and spectral methods. Overall, SFs suffer from their dependency to basic feature construction, with only Laplacian eigenfunctions working reliably. Notably, planning with SFs requires specific features: SFs can fail with generic encoder-type feature learning, even if the learned representation of states is reasonable (such as $(x, y)$). Forward-backward representations were best across the board, and provide reasonable zero-shot RL performance.

## ACKNOWLEDGEMENTS

The authors would like to thank Olivier Delalleau, Armand Joulin, Alessandro Lazaric, Sergey Levine, Matteo Pirotta, Andrea Tirinzoni, and the anonymous reviewers for helpful comments and questions on the research and manuscript.

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

## LIST OF APPENDICES

Appendix A sketches a proof of the basic properties of SFs and the FB representation, Theorems 1 and 2, respectively from Borsa et al. (2018) and Touati & Ollivier (2021).

Appendix B derives the loss (8) we use to learn $F$ and $B$, and the auxiliary orthonormalization loss.

Appendix C describes the precise mathematical relationship between the contrastive loss (17) and successor measures of the exploration policy.

Appendix D further discusses why low-rank models make more sense on successor measures than on the transition matrix $P$.

Appendix E proves that if $P$ is low-rank given by $\chi^\top \mu$, then successor features using $\varphi = \mu$ (but not $\chi$) provide optimal policies.

Appendix F proves that the loss (16) used to learn low-rank $P$ is an asymmetric version of the Laplacian eigenfunction loss (14).

Appendix G describes the detailed experimental setup: environments, architectures, policy training, methods for sampling of $z$, and hyperparameters. The full code can be found at
https://github.com/facebookresearch/controllable_agent

Appendix H contains the full table of experimental results, as well as aggregate plots over several variables (per task, per replay buffer, etc.).

Appendix I analyzes the learned features. We use feature rank analysis to study the degree of feature collapse for some methods. We also provide t-SNE visualizations of the learned embeddings for all methods (for the Maze environment).

Appendix J provides hyperparameter sensitivity plots (latent dimension $d$, learning rate, batch size, mixing ratio for $z$ sampling).

Appendix K describes a further baseline where we take a goal-oriented method inspired from Ma et al. (2020), and extend it to dense rewards by linearity. Since results were very poor except for Maze (which is goal-oriented), we did not discuss it in the main text.

Appendix L provides PyTorch snippets for the key losses, notably the **FB** loss, the **SF** loss as well as the various feature learning methods for **SF**.

# A    SKETCH OF PROOF OF THEOREMS 1 AND 2

To provide an intuition behind Theorems 1 and 2 on SFs and FB, we include here a sketch of proof in the finite state case. Full proofs in the general case can be found in the Appendix of Touati & Ollivier (2021).

For Theorem 1 (SFs), let us assume that the reward is linear in the features $\varphi$, namely, $r(s) = \varphi(s)^\top w$ for some $w \in \mathbb{R}^d$. Then by definition, $z_r = w$ since $z_r$ is the linear regression of the reward on the features (assuming features are linearly independent). Using the definition (4) of the successor features $\psi$, and taking the dot product with $z_r$, we obtain

$$\psi(s_0, a_0, z_r)^\top z_r = \mathbb{E}\left[\sum_t \gamma^t \varphi(s_{t+1})^\top z_r \mid s_0, a_0, \pi_{z_r}\right] = \mathbb{E}\left[\sum_t \gamma^t r(s_{t+1}) \mid s_0, a_0, \pi_{z_r}\right] \quad (20)$$

since $r = \varphi^\top w$. This means that $\psi(s_0, a_0, z_r)^\top z_r$ is the $Q$-function of reward $r$ for policy $\pi_{z_r}$. At the same time, by the definition (4), $\pi_{z_r}$ is defined as the argmax of $\psi(s_0, a_0, z_r)^\top z_r$. Therefore, the policy $\pi_{z_r}$ is the argmax of its own $Q$-function, meaning it is the optimal policy for reward $r$.

For Theorem 2 (FB), let us assume that FB perfectly satisfies the training criterion (5), namely, $M^{\pi_z} = F_z^\top B \operatorname{diag}(\rho)$ in matrix form. Thanks to the definition (1) of successor representations $M^\pi$, for any policy $\pi_z$, the $Q$-function for the reward $r$ can be written as $Q_r^{\pi_z} = M^{\pi_z} r$ in matrix form. This is equal to $F_z^\top B \operatorname{diag}(\rho) r$. Thus, if we define $z_r := B \operatorname{diag}(\rho) r = \mathbb{E}_{s \sim \rho}[B(s)r(s)]$, we obtain $Q_r^{\pi_z} = F_z^\top z_r$ for any $z \in \mathbb{R}^d$. In particular, the latter holds for $z = z_r$ as well: $F_{z_r}^\top z_r$ is the $Q$-function of $\pi_{z_r}$. Again, the policies $\pi_z$ are defined in (5) as the greedy policies of $F_z^\top z$, for any $z$. Therefore, $\pi_{z_r}$ is the argmax of its own $Q$-function. Hence, $\pi_{z_r}$ is the optimal policy for the reward $r$.

## B DERIVATION OF THE FORWARD-BACKWARD LOSS

Here we quickly derive the loss (8) used to train $F$ and $B$ such that $M^\pi(s, a, \mathrm{d}s') \approx F(s, a)^\top B(s') \rho(\mathrm{d}s')$. Training is based on the Bellman equation satisfied by $M^\pi$. This holds separately for each policy parameter $z$, so in this section we omit $z$ for simplicity.

Here $\rho(\mathrm{d}s')$ is the distribution of states in the dataset. Importantly, the resulting loss does not require to know this measure $\rho(\mathrm{d}s')$, only to be able to sample states $s \sim \rho$ from the dataset.

The successor measure $M^\pi$ satisfies a Bellman-like equation $M^\pi = P + \gamma P_\pi M^\pi$, as matrices in the finite case and as measures in the general case (Blier et al., 2021). We can learn FB by iteratively minimizing the Bellman residual $M^\pi - (P + \gamma P_\pi M^\pi)$ on the parametric model $M = F^\top B\rho$.

$M^\pi(s, a, \mathrm{d}s')$ is a measure on $s'$ for each $(s, a)$, so it is not obvious how to measure the size of the Bellman residual. In general, we can define a norm on such objects $M$ by taking the density with respect to the reference measure $\rho$,

$$\|M\|_\rho^2 := \mathbb{E}_{\substack{(s,a)\sim\rho \\ s'\sim\rho}} \left[ \left( \frac{M(s, a, \mathrm{d}s')}{\rho(\mathrm{d}s')} \right)^2 \right] \tag{21}$$

where $\frac{M(s,a,\mathrm{d}s')}{\rho(\mathrm{d}s')}$ is the density of $M$ with respect to $\rho$. [1] For finite states, $M$ is a matrix $M_{sas'}$ and this is just a $\rho$-weighted Frobenius matrix norm, $\|M\|_\rho^2 = \sum_{sas'} M_{sas'}^2 \rho(s, a)/\rho(s')$. (This is also how we proceed to learn a low-rank approximation of $P$ in (16).)

We define the loss on $F$ and $B$ as the norm of the Bellman residual on $M$ for the model $M = F^\top B\rho$. As usual in temporal difference learning, we use fixed, non-trainable target networks $\bar{F}$ and $\bar{B}$ for the right-hand-side of the Bellman equation. Thus, the Bellman residual is $F^\top B\rho - (P + \gamma P_\pi \bar{F}^\top \bar{B}\rho)$, and the loss is

$$\mathcal{L}(F, B) := \left\| F^\top B\rho - (P + \gamma P_\pi \bar{F}^\top \bar{B}\rho) \right\|_\rho^2 \tag{22}$$

When computing the norm $\|\cdot\|_\rho$, the denominator $\rho$ cancels out with the $F^\top B\rho$ terms, but we are left with a $P/\rho$ term. This term can still be integrated, because integrating $P(\mathrm{d}s'|s, a)/\rho(\mathrm{d}s')$ under $s' \sim \rho$ is equivalent to directly integrating under $s' \sim P(\mathrm{d}s'|s, a)$, namely, integrating under transitions $(s_t, a_t, s_{t+1})$ in the environment. This plays out as follows:

$$\mathcal{L}(F, B) = \mathbb{E}_{\substack{(s_t, a_t)\sim\rho \\ s'\sim\rho}} \left[ \left( F(s_t, a_t)^\top B(s') - \frac{P(\mathrm{d}s'|s_t, a_t)}{\rho(\mathrm{d}s')} - \gamma \mathbb{E}_{s_{t+1}\sim P(\mathrm{d}s_{t+1}|s_t, a_t)}[\bar{F}(s_{t+1}, \pi(s_{t+1}))^\top \bar{B}(s')] \right)^2 \right] \tag{23}$$

$$= \mathbb{E}_{\substack{(s_t, a_t, s_{t+1})\sim\rho \\ s'\sim\rho}} \left[ \left( F(s_t, a_t)^\top B(s') - \gamma \bar{F}(s_{t+1}, \pi(s_{t+1}))^\top \bar{B}(s') \right)^2 \right]$$
$$- 2 \mathbb{E}_{(s_t, a_t, s_{t+1})\sim\rho} \left[ F(s_t, a_t)^\top B(s_{t+1}) \right] + \texttt{Const} \tag{24}$$

where $\texttt{Const}$ is a constant term that we can discard since it does not depend on $F$ and $B$.

Apart from the discarded constant term, all terms in this final expression can be sampled from the dataset. Note that we have a $-2F(s_t, a_t)^\top B(s_{t+1})$ term where Touati & Ollivier (2021) have a $-2F(s_t, a_t)^\top B(s_t)$ term: this is because we define successor representations (1) using $s_{t+1}$ while Touati & Ollivier (2021) use $s_t$.

See also Appendix L for pseudocode (including the orthonormalization loss, and double networks for $Q$-learning as described in Appendix G).

---

[1] This is the dual norm on measures of the $L^2(\rho)$ norm on functions. It amounts to learning a model of $M(s, a, \mathrm{d}s')$ by learning relative densities to reach $s'$ knowing we start at $(s, a)$, relative to the average density $\rho(\mathrm{d}s')$ in the dataset.

**The orthonormalization loss.** An auxiliary loss is used to normalize $B$ so that $\mathbb{E}_{s\sim\rho}[B(s)B(s)^\top] \approx$ Id. This loss is

$$\mathcal{L}_{\mathrm{norm}}(B) := \left\|\mathbb{E}_\rho[BB^\top] - \mathrm{Id}\right\|_{\mathrm{Frobenius}}^2 \tag{25}$$

$$= \mathbb{E}_{s\sim\rho,\, s'\sim\rho}\left[(B(s)^\top B(s'))^2 - \|B(s)\|_2^2 - \|B(s')\|_2^2\right] + \texttt{Const.} \tag{26}$$

(The more complex expression in Touati & Ollivier (2021) has the same gradients up to a factor 4.)

**The auxiliary loss $\mathcal{L}'$ (9).** Learning $F(s,a,z)$ is equivalent to learning $F(s,a,z)^\top z'$ for all vectors $z'$. Yet the definition of the policies $\pi_z$ in FB only uses $F(s,a,z)^\top z$. Thus, as was done for SFs in Section 5.1, one may wonder if there is a scalar rather than vector loss to train $F$, that would reduce the error in the directions of $F$ used to define $\pi_z$.

In the case of FB, the full vector loss is needed to train $F$ and $B$. However, one can add the following auxiliary loss on $F$ to reduce the errors in the specific direction $F(s,a,z)^\top z$. This is obtained as follows.

Take the Bellman gap in the main FB loss (22): this Bellman gap is $F^\top B\rho - (P + \gamma P_\pi \bar{F}^\top \bar{B}\rho)$. To specialize this Bellman gap in the direction of $F(s,a,z)^\top z$, multiply by $B^\top(B\rho B^\top)^{-1}z$ on the right: this yields $F^\top z - (PB^\top(B\rho B^\top)^{-1}z + \gamma P_\pi \bar{F}^\top z)$ (using that we compute the loss at $\bar{B} = B$).

This new loss is the Bellman gap on $F^\top z$, with reward $PB^\top(B\rho B^\top)^{-1}z$.

This is the loss $\mathcal{L}'$ described in (9). It is a particular case of the main FB loss: $\mathcal{L} = 0$ implies $\mathcal{L}' = 0$, since we obtained $\mathcal{L}'$ by multiplying the Bellman gap of $\mathcal{L}$.

We use $\mathcal{L}'$ on top of the main FB loss to reduce errors in the direction $F(s,a,z)^\top z$. However, in the end, the differences are modest.

## C  RELATIONSHIP BETWEEN CONTRASTIVE LOSS AND SVD OF SUCCESSOR MEASURES

Here we prove the precise relationship between the contrastive loss (17) and SVDs of the successor measure of the exploration policy.

Intuitively, both methods push states together if they lie on the same trajectory, by increasing the dot product between the representations of $s_t$ and $s_{t+k}$. This is formalized as follows.

Let $\pi$ be the policy used to produce the trajectories in the dataset. Define

$$\tilde{M} := (1 - \gamma_{\mathtt{CL}}) \sum_{t \geq 0} \gamma_{\mathtt{CL}}^t P_\pi^{t+1} \tag{27}$$

where $\gamma_{\mathtt{CL}}$ is the parameter of the geometric distribution used to choose $k$ when sampling $s_t$ and $s_{t+k}$.

$\tilde{M}$ is a stochastic matrix in the discrete case, and a probability measure over $S$ in the general case: it is the normalized version of the successor measure (3) with $\pi$ the exploration policy.

By construction, the distribution of $s_{t+k}$ knowing $s_t$ with $k \sim \mathtt{Geom}(1 - \gamma_{\mathtt{CL}})$ is described by $\tilde{M}$. Therefore, we can rewrite the loss as

$$- \mathbb{E}_{\substack{k \sim \mathtt{Geom}(1 - \gamma_{\mathtt{CL}}) \\ (s_t, s_{t+k}) \sim \mathcal{D}}} \left[ \log \frac{\exp(\mathtt{cosine}(\varphi(s_t), \mu(s_{t+k})))}{\mathbb{E}_{s' \sim \mathcal{D}} \exp(\mathtt{cosine}(\varphi(s_t), \mu(s')))} \right] \tag{28}$$

$$= - \mathbb{E}_{s \sim \rho,\, s' \sim \rho} \left[ \frac{\tilde{M}(s, \mathrm{d}s')}{\rho(\mathrm{d}s')} \log \exp(\mathtt{cosine}(\varphi(s), \mu(s'))) \right]$$

$$+ \mathbb{E}_{s \sim \rho} \left[ \log \mathbb{E}_{s' \sim \mathcal{D}} \exp(\mathtt{cosine}(\varphi(s), \mu(s'))) \right] \tag{29}$$

Assume that $\varphi$ and $\mu$ are centered with unit norm, namely, $\|\varphi(s)\|_2 = 1$ and $\mathbb{E}_{s \sim \rho} \varphi(s) = 0$ and likewise for $\mu$. With unit norm, the cosine becomes just a dot product, and the loss is

$$\cdots = - \mathbb{E}_{s \sim \rho,\, s' \sim \rho} \left[ \frac{\tilde{M}(s, \mathrm{d}s')}{\rho(\mathrm{d}s')} \varphi(s)^\top \mu(s') \right] + \mathbb{E}_{s \sim \rho} \left[ \log \mathbb{E}_{s' \sim \mathcal{D}} \exp(\varphi(s)^\top \mu(s')) \right]. \tag{30}$$

A second-order Taylor expansion provides

$$\log \mathbb{E} \exp X = \mathbb{E} X + \tfrac{1}{2} \mathbb{E}[X^2] - \tfrac{1}{2} (\mathbb{E} X)^2 + O(|X|^3) \tag{31}$$

and therefore, with $\mathbb{E} \varphi = \mathbb{E} \mu = 0$, the loss is approximately

$$\cdots \approx - \mathbb{E}_{s \sim \rho,\, s' \sim \rho} \left[ \frac{\tilde{M}(s, \mathrm{d}s')}{\rho(\mathrm{d}s')} \varphi(s)^\top \mu(s') \right] + \tfrac{1}{2} \mathbb{E}_{s \sim \rho,\, s' \sim \rho} \left[ (\varphi(s)^\top \mu(s'))^2 \right] \tag{32}$$

$$= \tfrac{1}{2} \mathbb{E}_{s \sim \rho,\, s' \sim \rho} \left[ \left( \varphi(s)^\top \mu(s') - \frac{\tilde{M}(s, \mathrm{d}s')}{\rho(\mathrm{d}s')} \right)^2 \right] + \mathtt{Const} \tag{33}$$

where the constant term does not depend on $\varphi$ and $\mu$.

This is minimized when $\varphi^\top \mu$ is the SVD of $\tilde{M}/\rho$ in the $L^2(\rho)$ norm.

# D  DO FINITE-RANK MODELS ON THE TRANSITION MATRIX AND ON SUCCESSOR MEASURES MAKE SENSE?

It turns out finite-rank models are very different for $P$ or for $\sum \gamma^t P_\pi^t$. In typical situations, the spectrum of $P$ is concentrated around 1 while that of $\sum \gamma^t P_\pi^t$ is much more spread-out.

Despite the popularity of low-rank $P$ in theoretical RL works, $P$ is *never* close to low-rank in continuous-time systems: then $P$ is actually always close to the identity. Generally speaking, $P$ cannot be low-rank if most actions have a small effect. By definition, for any feature function $\varphi$, $(P_\pi \varphi)(s) = \mathbb{E}[\varphi(s_{t+1})|s_t = s]$. Intuitively, if actions have a small effect, then $s_{t+1}$ is close to $s_t$, and $\varphi(s_{t+1}) \approx \varphi(s_t)$ for continuous $\varphi$. This means that $P_\pi \varphi$ is close to $\varphi$, so that $P_\pi$ is close to the identity on a large subspace of feature functions $\varphi$. In the theory of continuous-time Markov processes, the time-$t$ transition kernel is given by $P_t = e^{tA}$ with $A$ the infinitesimal generator of the process (Levin et al., 2009, §20.1) (Øksendal, 1998, §8.1), hence $P_t$ is $\mathrm{Id} + O(t)$ for small timesteps $t$. In general, the transition matrix $P$ is better modeled as $\mathrm{Id} + \text{low-rank}$, which corresponds to a low-rank model of the Markov chain Laplacian $\mathrm{Id} - P$.

On the other hand, though $\sum \gamma^t P_\pi^t$ is never exactly low-rank (it is invertible), it has meaningful low-rank approximations under weak assumptions. For large $t$, $P_\pi^t$ becomes rank-one under weak assumptions (ergodicity), as it converges to the equilibrium distribution of the transition kernel. For large $\gamma$, the sum $\sum \gamma^t P_\pi^t$ is dominated by large $t$. Most eigenvalues of $P_\pi$ are close to 1, but taking powers $P_\pi^t$ sharpens the differences between eigenvalues: with $\gamma$ close to 1, going from $P_\pi$ to $\sum \gamma^t P_\pi^t = (\mathrm{Id} - \gamma P_\pi)^{-1}$ changes an eigenvalue $1 - \varepsilon$ into $1/\varepsilon$.

In short, on $P$ itself, there is little learning signal to differentiate between eigenvectors, but differences become visible over time. This may explain why FB works better than low-rank decompositions directly based on $P$ or the Laplacian.

For instance, consider the nearest-neighbor random walk on a length-$n$ cycle $\{0, 1, \ldots, n - 1 \mod n\}$, namely, moving in dimension 1. (This extends to any-dimensional grids.) The associated $P_\pi$ is not low-rank in any reasonable sense: the corresponding stochastic matrix is concentrated around the diagonal, and many eigenvalues are close to 1. Precisely, the eigenvalues are $\cos(2k\pi/n)$ with integer $k = \{0, \ldots, n/2\}$. This is $\approx 1 - 2\pi^2(k/n)^2$ when $k \ll n$. Half of the eigenvalues are between $\sqrt{2}/2$ and 1.

However, when $\gamma \to 1$, $\sum \gamma^t P_\pi^t$ has one eigenvalue $1/(1 - \gamma)$ and the other eigenvalues are $\frac{1}{1-\cos(2k\pi/n)} \approx n^2/2\pi^2 k^2$ with positive integer $k$: there is one large eigenvalue, then the others decrease like $\mathrm{cst}/k^2$. With such a spread-out spectrum, a finite-rank model makes sense.

# E   WHICH FEATURES ARE ANALOGOUS BETWEEN LOW-RANK $P$ AND SFs?

Here we explain the relationship between a low-rank model of $P$ and successor features. More precisely, if transition probabilities from $(s, a)$ to $s'$ can be written exactly as $\chi(s, a)^\top \mu(s')$, then SFs with basic features $\varphi := \mu$ will provide optimal policies for *any* reward function (Theorem 3).

Indeed, under the finite-rank model $P(\mathrm{d}s'|s, a) = \chi(s, a)^\top \mu(s')\rho(\mathrm{d}s')$, rewards only matter via the reward features $\mathbb{E}_{s' \sim \rho} \mu(s')r(s')$: namely, two rewards with the same reward features have the same $Q$-function, as the dynamics produces the same expected rewards. Then $Q$-functions are linear in these reward features, and using successor features with $\varphi := \mu$ provides the correct $Q$-functions, as follows.

**Theorem 3.** *Assume that* $P(\mathrm{d}s'|s, a) = \chi(s, a)^\top \mu(s')\rho(\mathrm{d}s')$. *Then successor features using the basic features* $\varphi := \mu$ *provide optimal policies for any reward function.*

This is why we use $\mu$ rather than $\chi$ for the SF basic features. This is also why we avoided the traditional notation $P(s'|s, a) = \varphi(s, a)^\top \mu(s')$ often used for low-rank $P$, which induces a conflict of notation with the $\varphi$ in SFs, and suggests the wrong analogy.

Meanwhile, $\chi$ plays a role more analogous to SFs' $\psi$, although for one-step transitions instead of multistep transitions as in SFs: $Q$-functions are linear combinations of the features $\chi$. In particular, the optimal $Q$-function for reward $r$ is $Q_r^\star = \chi^\top w_r$ for some $w_r$. But contrary to successor features, there is no simple correspondence to compute $w_r$ from $r$.

*Proof.* Let $\pi$ be any policy. On a finite space in matrix notation, and omitting $\rho$ for simplicity, the assumption $P = \chi^\top \mu$ implies $P_\pi = \chi_\pi^\top \mu$ where $\chi_\pi(s) := \mathbb{E}_{a \sim \pi(s)} \chi(s, a)$ are the $\pi$-averaged features. Then,

$$Q_r^\pi = P \sum_{t \geq 0} \gamma^t P_\pi^t r \tag{34}$$

$$= \chi^\top \mu \sum_{t \geq 0} \gamma^t \left(\chi_\pi^\top \mu\right)^t r \tag{35}$$

$$= \chi^\top \left(\sum_{t \geq 0} \gamma^t \left(\mu \chi_\pi^\top\right)^t\right) \mu r. \tag{36}$$

Thus, $Q$-functions are expressed as $Q_r^\pi(s, a) = \chi(s, a)^\top w(\pi, r)$ with $w(\pi, r) = \left(\sum_{t \geq 0} \gamma^t (\mu \chi_\pi^\top)^t\right) \mu r$.

Moreover, rewards only matter via $\mu r$. Namely, two rewards with the same $\mu r$ have the same $Q$-function for every policy.

In full generality on continuous spaces with $\rho$ again, the same holds with $\mu(s)\rho(\mathrm{d}s)$ instead of $\mu(s)$, and $\mathbb{E}_{s \sim \rho} \mu(s)r(s)$ instead of $\mu r$.

Now, let $r$ be any reward function, and let $r'$ be its $L^2(\rho)$-orthogonal projection onto the space generated by the features $\mu$. By construction, $r - r'$ is $L^2(\rho)$-orthogonal to $\mu$, namely, $\mathbb{E}_\rho \mu(r - r') = 0$. So $\mathbb{E}_\rho \mu r = \mathbb{E}_\rho \mu r'$. Therefore, by the above, $r$ and $r'$ have the same $Q$-function for every policy.

By definition, $r'$ lies in the linear span of the features $\mu$. By Theorem 1, SFs with features $\varphi = \mu$ will provide optimal policies for $r'$. Since $r$ and $r'$ have the same $Q$-function for every policy, an optimal policy for $r'$ is also optimal for $r$. $\qquad\square$

# F  RELATIONSHIP BETWEEN LAPLACIAN EIGENFUNCTIONS AND LOW-RANK $P$ LEARNING

The loss (16) used to learn a low-rank model of the transition probabilities $P$ is an asymmetric version of the Laplacian eigenfunction loss (14) with $\lambda = 1$.

Said equivalently, if we use the low-rank $P$ loss (16) constrained with $\chi = \mu$ to learn a low-rank model of $P_\pi$ instead of $P$ (with $\pi$ the exploration policy), then we get the Laplacian eigenfunction loss (14) with $\lambda = 1$.

Indeed, set $\lambda = 1$ in (14). Assume that the distributions of $s_t$ and $s_{t+1}$ in the dataset are identical on average (this happens, e.g., if the dataset is made of long trajectories or if $\rho$ is close enough to the invariant distribution of the exploration policy). Then, in the Laplacian loss (14), the norms from the first term cancel those from the second, and the Laplacian loss simplifies to

$$(14) = \mathbb{E}_{(s_t,s_{t+1})\sim\mathcal{D}}\left[\|\varphi(s_t) - \varphi(s_{t+1})\|^2\right] + \mathbb{E}_{\substack{s\sim\mathcal{D}\\s'\sim\mathcal{D}}}\left[(\varphi(s)^\top\varphi(s'))^2 - \|\varphi(s)\|_2^2 - \|\varphi(s')\|_2^2\right] \tag{37}$$

$$= \mathbb{E}_{s_t\sim\mathcal{D}}\|\varphi(s_t)\|^2 + \mathbb{E}_{s_{t+1}\sim\mathcal{D}}\|\varphi(s_t)\|^2 - 2\,\mathbb{E}_{(s_t,s_{t+1})\sim\mathcal{D}}\left[\varphi(s_t)^\top\varphi(s_{t+1})\right] \tag{38}$$

$$+ \mathbb{E}_{\substack{s\sim\mathcal{D}\\s'\sim\mathcal{D}}}\left[(\varphi(s)^\top\varphi(s'))^2 - \|\varphi(s)\|_2^2 - \|\varphi(s')\|_2^2\right] \tag{39}$$

$$= -2\,\mathbb{E}_{(s_t,s_{t+1})\sim\mathcal{D}}\left[\varphi(s_t)^\top\varphi(s_{t+1})\right] + \mathbb{E}_{\substack{s\sim\mathcal{D}\\s'\sim\mathcal{D}}}\left[(\varphi(s)^\top\varphi(s'))^2\right]. \tag{40}$$

This is the same as the low-rank loss (16) if we omit actions $a$ and constrain $\chi = \mu$.

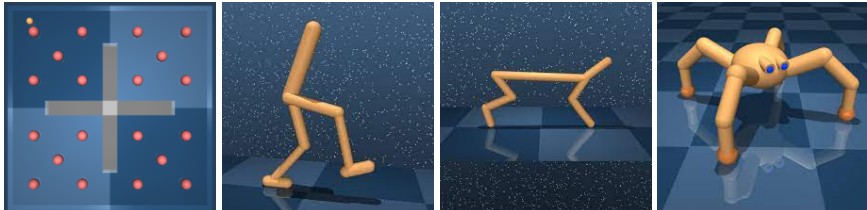

Figure 3: Maze, Walker, Cheetah and Quadruped environments used in our experiments. In the Mmaze domain (left), we show an example of an initial state (yellow point) and the 20 test goals (red circles).

## G  EXPERIMENTAL SETUP

In this section we provide additional information about our experiments.

Code snippets for the main losses are given in Appendix L. The full code can be found at
https://github.com/facebookresearch/controllable_agent

### G.1  ENVIRONMENTS

All the environments considered in this paper are based on the *DeepMind Control Suite* (Tassa et al., 2018).

- **Point-mass Maze**: a 2-dimensional continuous maze with four rooms. The states are 4-dimensional vectors consisting of positions and velocities of the point mass $(x, y, v_x, x_y)$, and the actions are 2-dimensional vectors. At test, we assess the performance of the agents on 20 goal-reaching tasks (5 goals in each room described by their $(x, y)$ coordinates).
- **Walker**: a planar walker. States are 24-dimensional vectors consisting of positions and velocities of robot joints, and actions are 6-dimensional vectors. We consider 4 different tasks at test time: `walker_stand` reward is a combination of terms encouraging an upright torso and some minimal torso height, while `walker_walk` and `walker_run` rewards include a component encouraging some minimal forward velocity. `walker_flip` reward includes a component encouraging some mininal angular momentum.
- **Cheetah**: a running planar biped. States are 17-dimensional vectors consisting of positions and velocities of robot joints, and actions are 6-dimensional vectors. We consider 4 different tasks at test time: `cheetah_walk` and `cheetah_run` rewards are linearly proportional to the forward velocity up to some desired values: 2 m/s for `walk` and 10 m/s for `run`. Similarly, `walker_walk_backward` and `walker_run_backward` rewards encourage reaching some minimal backward velocities.
- **Quadruped**: a four-leg ant navigating in 3D space. States and actions are 78-dimensional and 12-dimensional vectors, respectively. We consider 4 tasks at test time: `quadruped_stand` reward encourages an upright torso. `quadruped_walk` and `quadruped_run` include a term encouraging some minimal torso velecities. `quadruped_walk` includes a term encouraging some minimal height of the center of mass.

### G.2  ARCHITECTURES

We use the same architectures for all methods.

- The backward representation network $B(s)$ and the feature network $\varphi(s)$ are represented by a feedforward neural network with three hidden layers, each with 256 units, that takes as input a state and outputs a L2-normalized embedding of radius $\sqrt{d}$.
- For both successor features $\psi(s, a, z)$ and forward network $F(s, a, z)$, we first preprocess separately $(s, a)$ and $(s, z)$ by two feedforward networks with two hidden layers (each with 1024 units) to 512-dimentional space. Then we concatenate their two outputs and pass it

  into another 2-layer feedforward network (each with 1024 units) to output a $d$-dimensional vector.
  - For the policy network $\pi(s, z)$, we first preprocess separately $s$ and $(s, z)$ by two feedforward networks with two hidden layers (each with 1024 units) to 512-dimentional space. Then we concatenate their two outputs and pass it into another 2-layer feedforward network (each with 1024 units) to output to output a $d_A$-dimensional vector, then we apply a `Tanh` activation as the action space is $[-1, 1]^{d_A}$.

For all the architectures, we apply a layer normalization (Ba et al., 2016) and `Tanh` activation in the first layer in order to standardize the states and actions. We use `Relu` for the rest of layers. We also pre-normalized $z$: $z \leftarrow \sqrt{d}\frac{z}{\|z\|_2}$ in the input of $F$, $\pi$ and $\psi$. Empirically, we observed that removing preprocessing and pass directly a concatenation of $(s, a, z)$ directly to the network leads to unstable training. The same holds when we preprocess $(s, a)$ and $z$ instead of $(s, a)$ and $(s, z)$, which means that the preprocessing of $z$ should be influenced by the current state.

For maze environments, we added an additional hidden layer after the preprocessing (for both policy and forward / successor features) as it helped to improve the results.

### G.3 SAMPLING OF $z$

We mix two methods for sampling $z$:

  1. We sample $z$ uniformly in the sphere of radius $\sqrt{d}$ in $\mathbb{R}^d$ (so each component of $z$ is of size $\approx 1$).

  2. We sample $z$ using the formula for $z_r$ corresponding to the reward for reaching a random goal state $s$ in Theorems 1–2. Namely, we set $z = B(s)$ for FB and $z = \left(\sum_{i=1}^m \varphi(s_i)\varphi(s_i)^\top\right)^+ \varphi(s)$ for SFs, where $s \sim \rho$ is a random state sampled from the replay buffer and $s_i$ are states in a minibatch.

For the main series of results, we used a $50\%$ mix ratio for those two methods. Different algorithms can benefit from different ratios: this is explored in Appendix

### G.4 LEARNING THE POLICIES $\pi_z$: POLICY NETWORK

As the action space is continuous, we could not compute the $\arg\max$ over action in closed form. Instead, we consider a latent-conditioned policy network $\pi_\eta : S \times Z \to A$, and we learn the policy parameters $\eta$ by performing stochastic gradient ascent on the objective $\mathbb{E}_{s,z}[F(s, \pi_\eta(s), z)^\top z]$ for FB or $\mathbb{E}_{s,z}[\psi(s, \pi_\eta(s), z)^\top z]$ for SFs.

We also incorporate techniques introduced in the TD3 paper (Fujimoto et al., 2018) to address function approximation error in actor-critic methods: double networks and target policy smoothing, adding noise $\varepsilon$ to the actions.

Let $\theta_1$ and $\theta_2$ the parameters of two forward networks and let $\omega$ the parameters of the backward network. Let $\theta_1^-$, $\theta_2^-$ and $\omega^-$ be the parameters of their corresponding target networks.

Let $\{(s_i, a_i, s_i^{\text{next}})\}_{i \in I} \subset \mathcal{D}$ a mini-batch of size $|I| = b$ of transitions and let $\{z_i\}_{i \in I}$ a mini-batch of size $|I| = b$ of latent variables sampled according to G.3. The empirical version of the main FB loss in (8) is (with an additional $1/2$ factor):

$$\mathcal{L}(\theta_k, \omega) = \frac{1}{2b(b-1)} \sum_{\substack{i,j \in I^2 \\ i \neq j}} \left( F_{\theta_k}(s_i, a_i, z_i)^\top B_\omega(s_j^{\text{next}}) - \gamma \min_{l=1,2} F_{\theta_l^-}(s_i^{\text{next}}, \pi_\eta(s_i^{\text{next}}) + \varepsilon_i, z_i)^\top B_{\omega^-}(s_j^{\text{next}}) \right)^2$$

$$- \frac{1}{b} \sum_{i \in I} F_{\theta_k}(s_i, a_i, z_i)^\top B_\omega(s_i^{\text{next}}) \quad \forall k = 1, 2 \tag{41}$$

where $\varepsilon_i$ is sampled from a truncated centered Gaussian with variance $\sigma^2$ (for policy smoothing). The empirical version of the auxiliary $F$ loss in (9) is: for $k = 1, 2$,

$$\mathscr{L}'(\theta_k) = \frac{1}{b} \sum_{i \in I} \left( F_{\theta_k}(s_i, a_i, z_i)^\top z_i - B_\omega(s_i^{\text{next}})^\top \text{Cov}^+ z_i - \gamma \min_{l=1,2} F_{\theta_l^-}(s_i^{\text{next}}, \pi_\eta(s_i^{\text{next}}, z) + \varepsilon_i, z_i)^\top z_i \right)^2$$

(42)

where $\text{Cov}^+$ is the pseudo-inverse of the empirical covariane matrix $\text{Cov} = \frac{1}{b} \sum_{i \in I} B_\omega(s_i) B_\omega(s_i)^\top$. We use $1/d$ as regularization coefficient in front of $\mathscr{L}'(\theta_k)$.

For policy training, the empirical loss is:

$$\mathscr{L}(\eta) = -\frac{1}{b} \sum_{i \in I} \min_{l=1,2} F_{\theta_l}(s_i, \pi_\eta(s_i, z) + \varepsilon_i, z_i)^\top z_i$$

(43)

The same techniques are also used for SFs.

### G.5 HYPERPARAMETERS

Table 1 summarizes the hyperparameters used in our experiments.

A hyperparameter sensitivity analysis for two domains (Walker and Cheetah) is included in Appendix J.

Table 1: Hyperparameters used in our experiments.

| Hyperparameter | Value |
|---|---|
| Replay buffer size | $5 \times 10^6$ ($10 \times 10^6$ for maze) |
| Representation dimension | 50 (100 for maze) |
| Batch size | 1024 |
| Discount factor $\gamma$ | 0.98 (0.99 for maze) |
| Optimizer | Adam |
| Learning rate | $10^{-4}$ |
| Mixing ratio for $z$ sampling | 0.5 |
| Momentum coefficient for target networks | 0.99 |
| Stddev $\sigma$ for policy smoothing | 0.2 |
| Truncation level for policy smoothing | 0.3 |
| Number of gradient steps | $10^6$ |
| Number of reward labels for task inference | $10^4$ |
| Discount factor $\gamma_{\texttt{CL}}$ for $\texttt{CL}$ | 0.6 (0.2 for maze) |
| Regularization weight for orthonormality loss (spectral methods) | 1 |

**Hyperparameter tuning.** Since all methods share the same core, we chose to use the same hyperparameters rather than tune per method, which could risk leading to less robust conclusions.

We did not do hyperparameter sweeps for each baseline and task, first because this would have been too intensive given the number of setups, and, second, this would be too close to going back to a supervised method for each task.

Instead, to avoid any overfitting, we tuned architectures and hyperparameters by hand on the Walker environment only, with the RND replay buffer, and reused these parameters across all methods and tasks (except for Maze, on which all methods behave differently). We identified some trends by monitoring learning curves and downstream performance on Walker, and we fixed a configuration that led to overall good performance for all the methods. We avoided a full sweep to avoid overfitting based on Walker, to focus on robustness.

For instance, the learning rate $10^{-4}$ seemed to work well with *all* methods, as seen in Appendix J.

Some trends were common between all methods: indeed, all the methods share a common core (training of the successor features $\psi$ or $F$, and training of the policies $\pi$) and differ by the training of the basic features $\varphi$ or $B$.

For the basic features $\varphi$, the various representation learning losses were easy to fit and had low hyperparameter sensitivity: we always observed smooth decreasing of losses until convergence. The challenging part was learning $\psi$ and $F$ and their corresponding policies, which is common across methods.

# H   DETAILED EXPERIMENTAL RESULTS

We report the full experimental results, first as a table (Section H.1).

In Section H.2 we plot aggregated results for easier interpretation: first, aggregated across all tasks with a plot of the variability for each method; second, aggregated over environments (since each environment corresponds to a trained zero-shot model); third, for each individual task but still aggregated over replay buffers.

In Section H.3 we plot all individual results per task and replay buffer.

In Section H.4 we plot aggregate results split by replay buffer.

## H.1   FULL TABLE OF RESULTS

| Buffer | Domain | Task | Rand | AEnc | ICM | Latent | Trans | Method Lap | LRA-P | CL | LRA-SR | FB | APS |
|---|---|---|---|---|---|---|---|---|---|---|---|---|---|
| APS | cheetah | run | 145±7 | 97±3 | 432±8 | 49±14 | 133±15 | 198±4 | 8±1 | 2±0 | 247±10 | 267±33 | 25±3 |
| | | run-backward | 189±20 | 365±6 | 404±3 | 32±3 | 382±3 | 221±4 | 1±0 | 14±6 | 261±5 | 238±7 | 98±21 |
| | | walk | 665±60 | 404±19 | 928±54 | 302±67 | 287±53 | 900±49 | 75±31 | 2±1 | 918±22 | 844±51 | 144±11 |
| | | walk-backward | 653±75 | 982±0 | 986±0 | 453±105 | 985±0 | 937±17 | 8±1 | 150±51 | 983±0 | 981±1 | 452±77 |
| | maze | reach | 11±5 | 5±1 | 8±3 | 10±4 | 15±5 | 432±18 | 436±16 | 12±3 | 145±8 | 410±16 | 59±7 |
| | quadruped | jump | 784±5 | 727±12 | 164±20 | 554±24 | 624±14 | 718±18 | 309±50 | 102±29 | 632±20 | 649±23 | 311±24 |
| | | run | 487±1 | 459±5 | 91±19 | 382±14 | 411±16 | 491±3 | 238±21 | 53±15 | 448±6 | 476±8 | 196±10 |
| | | stand | 966±3 | 925±16 | 248±46 | 752±36 | 890±17 | 963±1 | 497±53 | 56±9 | 872±20 | 924±13 | 417±10 |
| | | walk | 543±19 | 444±8 | 108±25 | 489±20 | 490±20 | 524±13 | 228±26 | 45±14 | 463±18 | 712±29 | 205±6 |
| | walker | flip | 158±10 | 317±31 | 452±3 | 299±53 | 471±15 | 454±12 | 340±18 | 69±26 | 186±21 | 413±16 | 39±2 |
| | | run | 96±4 | 127±8 | 290±9 | 359±11 | 263±12 | 289±10 | 115±11 | 63±11 | 204±26 | 346±14 | 35±3 |
| | | stand | 486±27 | 617±37 | 925±19 | 868±57 | 864±24 | 895±9 | 643±35 | 205±51 | 591±35 | 822±26 | 176±17 |
| | | walk | 177±30 | 462±58 | 724±41 | 857±8 | 816±30 | 386±40 | 159±20 | 76±44 | 671±19 | 817±15 | 34±2 |
| Proto | cheetah | run | 58±8 | 84±10 | 333±14 | 27±5 | 322±5 | 142±2 | 149±3 | 0±0 | 209±10 | 210±13 | - |
| | | run-backward | 149±6 | 262±7 | 329±3 | 30±5 | 274±7 | 146±2 | 133±6 | 16±13 | 230±6 | 157±7 | - |
| | | walk | 287±38 | 327±51 | 961±24 | 112±11 | 929±13 | 722±19 | 770±31 | 1±0 | 860±31 | 908±18 | - |
| | | walk-backward | 664±39 | 973±3 | 987±0 | 101±15 | 982±0 | 798±16 | 629±31 | 151±86 | 979±1 | 742±46 | - |
| | maze | reach | 27±5 | 7±1 | 7±2 | 15±4 | 8±1 | 571±16 | 556±15 | 19±5 | 134±7 | 326±16 | - |
| | quadruped | jump | 196±29 | 185±37 | 137±27 | 209±32 | 282±27 | 177±26 | 184±25 | 57±14 | 113±12 | 183±24 | - |
| | | run | 134±17 | 234±8 | 88±13 | 123±17 | 191±14 | 125±14 | 166±19 | 65±16 | 99±9 | 137±14 | - |
| | | stand | 413±56 | 321±42 | 220±29 | 270±38 | 436±34 | 231±52 | 264±34 | 86±12 | 215±47 | 287±53 | - |
| | | walk | 148±21 | 171±21 | 122±10 | 156±24 | 212±12 | 135±16 | 172±21 | 40±13 | 100±26 | 280±52 | - |
| | walker | flip | 133±12 | 346±6 | 510±13 | 443±30 | 456±10 | 548±13 | 281±22 | 78±21 | 551±13 | 507±18 | - |
| | | run | 84±2 | 234±9 | 259±18 | 347±20 | 303±9 | 280±22 | 183±28 | 45±5 | 391±15 | 336±9 | - |
| | | stand | 415±26 | 905±9 | 910±13 | 582±62 | 951±5 | 937±5 | 687±41 | 253±49 | 874±23 | 902±25 | - |
| | | walk | 125±21 | 632±33 | 839±17 | 791±17 | 832±16 | 883±30 | 300±31 | 92±16 | 867±12 | 917±7 | - |
| RND | cheetah | run | 64±3 | 68±5 | 96±8 | 183±26 | 66±4 | 50±5 | 6±1 | 163±14 | 138±17 | 247±9 | - |
| | | run-backward | 96±6 | 162±18 | 160±22 | 60±4 | 143±17 | 90±7 | 2±0 | 124±13 | 82±8 | 185±17 | - |
| | | walk | 289±22 | 337±28 | 401±40 | 567±70 | 286±30 | 330±65 | 29±11 | 622±28 | 446±36 | 827±41 | - |
| | | walk-backward | 469±38 | 542±70 | 743±60 | 345±59 | 572±100 | 499±40 | 14±1 | 517±60 | 352±45 | 793±66 | - |
| | maze | reach | 9±2 | 4±1 | 4±1 | 8±4 | 8±4 | 707±12 | 759±7 | 736±3 | 532±20 | 710±8 | - |
| | quadruped | jump | 770±7 | 474±40 | 176±13 | 663±20 | 806±14 | 490±48 | 447±31 | 326±72 | 731±9 | 651±8 | - |
| | | run | 465±4 | 415±10 | 98±13 | 418±10 | 478±8 | 399±26 | 301±12 | 263±40 | 461±6 | 429±3 | - |
| | | stand | 919±19 | 770±28 | 426±45 | 830±31 | 973±2 | 720±27 | 552±32 | 529±70 | 944±11 | 815±2 | - |
| | | walk | 586±28 | 486±37 | 90±13 | 527±21 | 552±33 | 410±30 | 310±14 | 210±42 | 516±30 | 528±10 | - |
| | walker | flip | 267±28 | 332±18 | 461±9 | 34±3 | 399±19 | 569±30 | 512±27 | 50±9 | 454±23 | 578±10 | - |
| | | run | 96±8 | 167±12 | 251±11 | 47±22 | 250±6 | 299±20 | 325±10 | 38±5 | 350±14 | 388±8 | - |
| | | stand | 516±36 | 733±22 | 813±10 | 191±62 | 853±18 | 836±38 | 904±21 | 326±41 | 828±20 | 890±15 | - |
| | | walk | 152±35 | 457±33 | 518±74 | 30±2 | 607±31 | 748±75 | 818±36 | 53±10 | 853±15 | 760±19 | - |

Table 2: Score of each method, split by task and replay buffer. Average over ten random seeds, with ±1σ estimated standard deviation on this average estimator. For Maze, we report the average over the 20 goals defined in the environment. We highlight the three leading methods (four when confidence intervals overlap) for each task.

## H.2 Aggregate Plots of Results

Here we plot the results, first averaged over everything, then by environment averaged over the tasks of that environment, and finally by task.

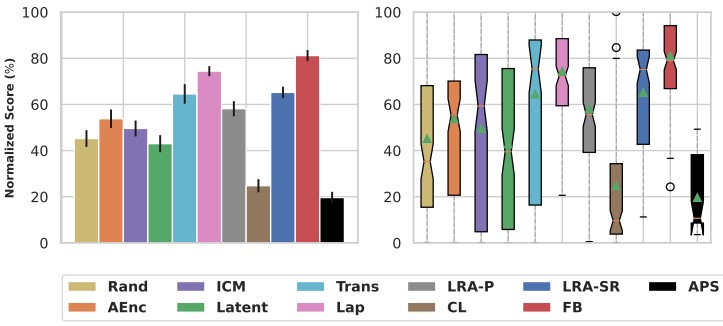

Figure 4: Zero-shot scores of ten SF methods and FB, aggregated over tasks using normalized scores as described in the text. To assess variability, the box plot on the right shows the variations of the distribution of normalized scores over random seeds, environments, and replay buffers.

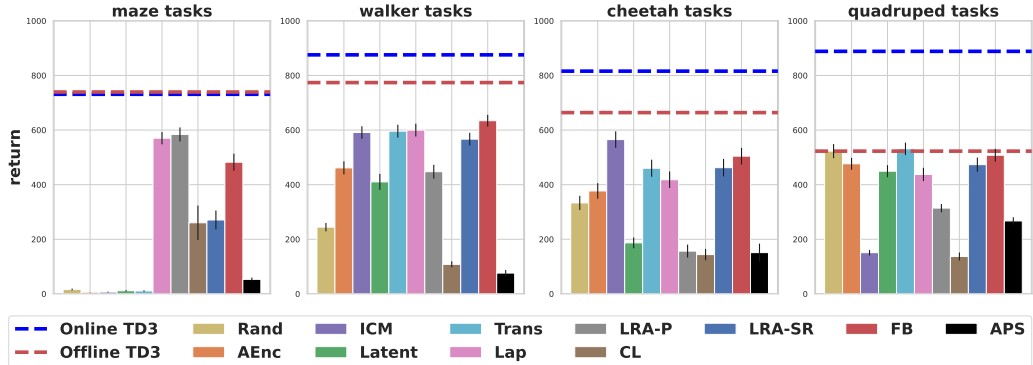

Figure 5: Zero-shot scores averaged over tasks for each environment, with supervised online and offline TD3 as toplines. Average over 3 replay buffers and 10 random seeds.

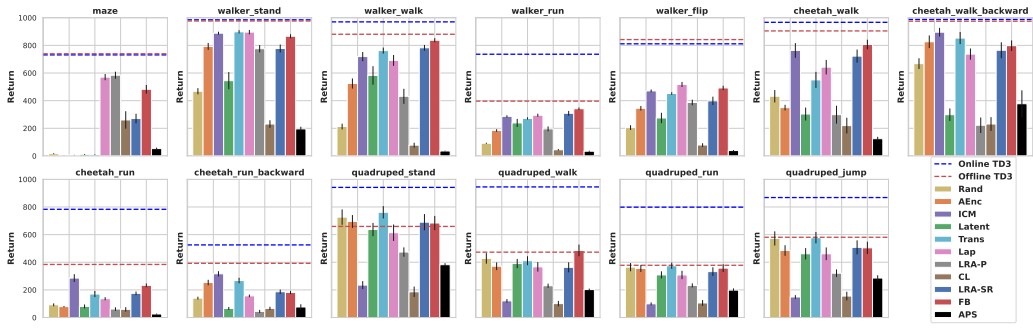

Figure 6: Zero-shot scores for each task, with supervised online and offline TD3 as toplines. Average over 3 replay buffers and 10 random seeds.

### H.3 FULL PLOTS OF RESULTS PER TASK AND REPLAY BUFFER

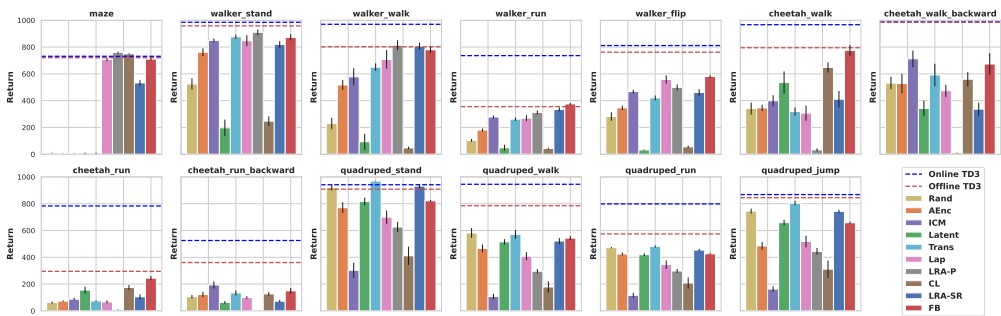

Figure 7: Per-task results on the RND replay buffer, average over 10 random seeds.

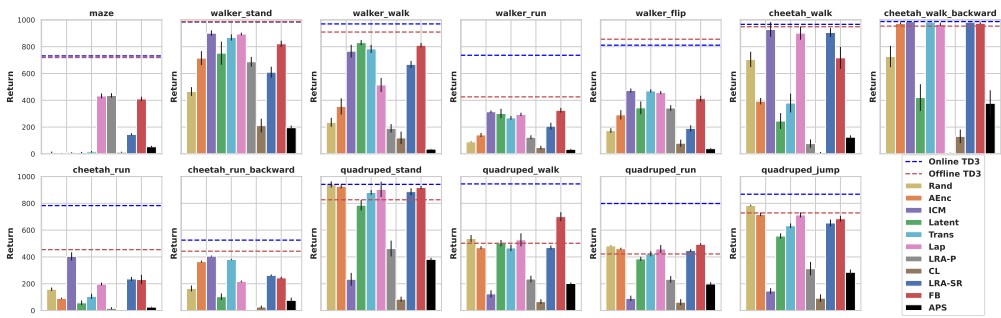

Figure 8: Per-task results on the APS replay buffer, average over 10 random seeds.

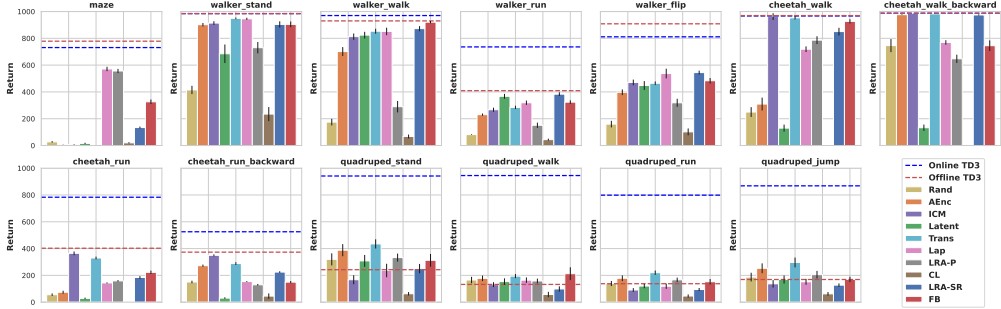

Figure 9: Per-task results on the Proto replay buffer, average over 10 random seeds.

### H.4 INFLUENCE OF THE REPLAY BUFFER

Here we plot the influence of the replay buffer, by reporting results separated by replay buffer, but averaged over the tasks corresponding to each environment (Fig 10).

Overall, there is a clear failure case of the Proto buffer on the Quadruped environment: the TD3 supervised baseline performs poorly for all tasks in that environment.

Otherwise, results are broadly consistent on the different replay buffers: with a few exceptions, the same methods succeed or fail on the same environments.

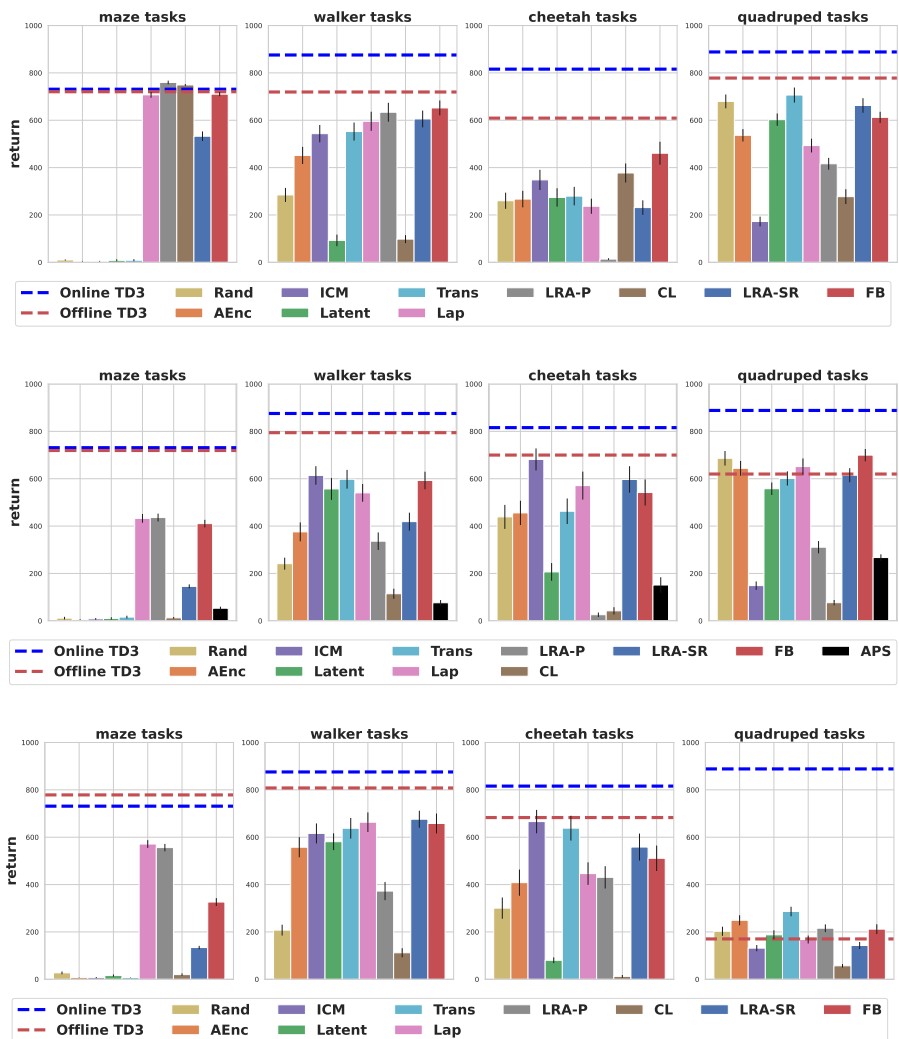

Figure 10: Results on each replay buffer: RND (top), APS (middle), Proto (bottom). Average over 4 tasks for the Walker, Cheetah and Quadruped environments, average over 20 goals for Maze.

In Fig. 11, we plot results aggregated over all tasks but split by replay buffer. Box plots further show variability within random seeds and environments for a given replay buffer.

Overall method rankings are broadly consistent between RND and APS, except for **CL**. Note that the aggregated normalized score for Proto is largely influenced by the failure on Quadruped: normalization by a very low baseline (Fig. 10, Quadruped plot for Proto), somewhat artificially pushes Trans high up (this shows the limit of using scores normalized by offline TD3 score), while the other methods' rankings are more similar to RND and APS.

**FB** and **Lap** work very well in all replay buffers, with **LRA-SR** and **Trans** a bit behind due to their feailures on some tasks.

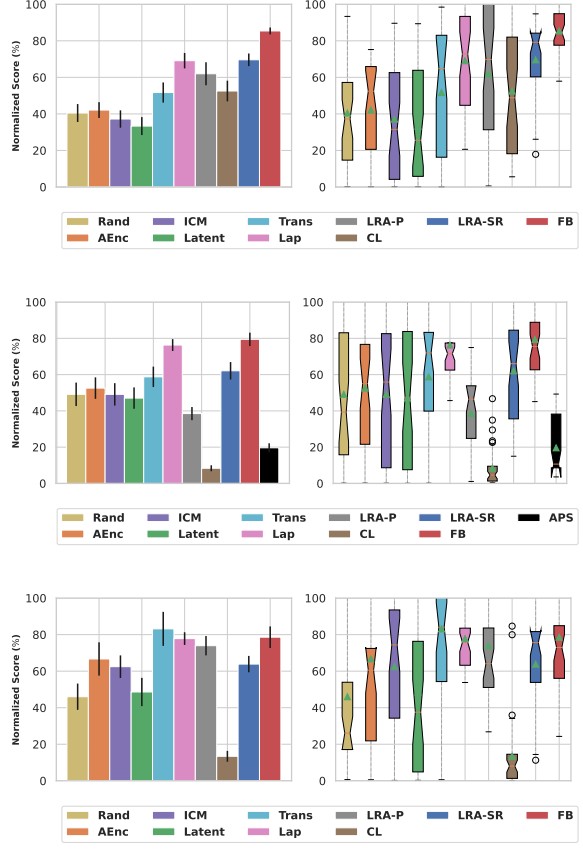

Figure 11: Zero-shot scores by replay buffer, as a percentage of the supervised score of offline TD3 trained on the same buffer, averaged over tasks and environments and random seeds. Top: RND buffer; middle: APS buffer; bottom: Proto buffer.

# I  ANALYSIS OF THE LEARNED FEATURES

## I.1  FEATURE RANK

Here we test the hypothesis that feature collapse for some methods is responsible for some cases of bad performance. This is especially relevant for Maze, where some methods may have little incentive to learn more features beyond the original two features $(x, y)$.

We report in Table 3 the effective rank of learned features $B$ or $\varphi$: this is computed as in (Lyle et al., 2022), as the fraction of eigenvalues of $\mathbb{E}_{s \sim \rho} B(s)B(s)^\top$ or $\mathbb{E}_{s \sim \rho} \varphi(s)\varphi(s)^\top$ above a certain threshold.

| Domain | | | | | | Method | | | | |
|---|---|---|---|---|---|---|---|---|---|---|
| | Rand | AEnc | ICM | Latent | Trans | Lap | LRA-P | CL | LRA-SR | FB |
| Maze | 0.38 | 0.31 | 0.33 | 0.32 | 0.15 | 1.0 | 1.0 | 1.0 | 1.0 | 1.0 |
| Walker | 1,0 | 0.91 | 1.0 | 1.0 | 0.90 | 1.0 | 1.0 | 0.65 | 1.0 | 1.0 |
| Cheetah | 1.0 | 0.96 | 1.0 | 1.0 | 1.0 | 1.0 | 1.0 | 1.0 | 1.0 | 1.0 |
| Quadruped | 1.0 | 0.98 | 0.30 | 1.0 | 0.15 | 1.0 | 0.84 | 1.0 | 1.0 | 1.0 |

Table 3: Feature rank (Lyle et al., 2022) of $\varphi$ or $B$ for each method, computed as $\frac{1}{d} \# \left\{ \sigma \in \texttt{eig}\left( \frac{1}{n} \sum_{i=1}^{n} \varphi(s_i)\varphi(s_i)^\top \right) \mid \sigma > \varepsilon \right\}$, trained on RND replay buffer and averaged over 10 random seeds. We use $n = 100,000$ samples to estimate the covariance, and $\varepsilon = 10^{-4}$.

For most methods and all environments except Maze, more than $90\%$ (often $100\%$) of eigenvalues are above $10^{-4}$, so the effective rank is close to full.

The Maze environment is a clear exception: on Maze, for the **Rand**, **AEnc**, **Trans**, **Latent** and **ICM** methods, only about one third of the eigenvalues are above $10^{-4}$. The other methods keep $100\%$ of the eigenvalues above $10^{-4}$. This is perfectly aligned with the performance of each method on Maze.

So rank reduction does happen for some methods. This reflects the fact that two features $(x, y)$ already convey the necessary information about states and dynamics, but are not sufficient to solve the problem via successor features. In the Maze environment, auto-encoder or transition models can perfectly optimize their loss just by keeping the original two features $(x, y)$, and they have no incentive to learn other features, so the effective rank could have been 2.

These methods may benefit from auxiliary losses to prevent eigenvalue collapse, similar to the orthonormalization loss used for $B$. We did not include such losses, because we wanted to keep the same methods (autoencoders, ICM, transition model...) used in the literature. But even with an auxiliary loss, keeping a full rank could be achieved just by keeping $(x, y)$ and then blowing up some additional irrelevant features.

## I.2 EMBEDDING VISUALIZATION VIA T-SNE

We visualize the learned state feature embeddings for Maze by projecting them into 2-dimensional space using t-SNE (Van der Maaten & Hinton, 2008) in Fig. I.2.

**AEnc**, **Trans**, **FB**, **LRA-P** and **CL** all recover a picture of the maze. For the other methods the picture is less clear (notably, **Rand** gets a near-circle, possibly as an instance of concentration of measure theorems).

Whether t-SNE preserves the shape of the maze does not appear to be correlated to performance: **AEnc** and **Trans** learn nice features but perform poorly, while the t-SNE of **Lap** is not visually interpretable but performance is good.

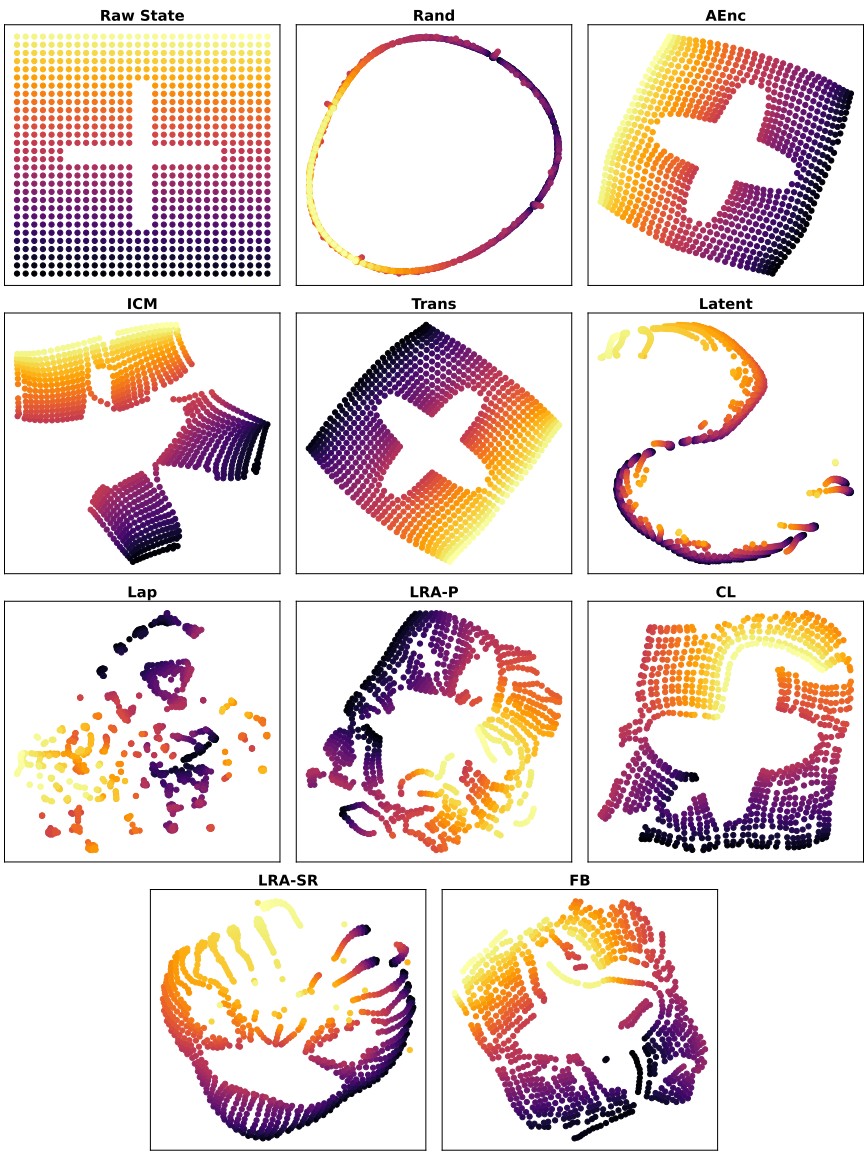

Figure 12: Visualization of embedding vectors obtained by each method ($\varphi$ for **SF** and $B$ for **FB**) on the maze domain after projecting them in two-dimensional space with t-SNE .

# J HYPERPARAMETERS SENSITIVITY

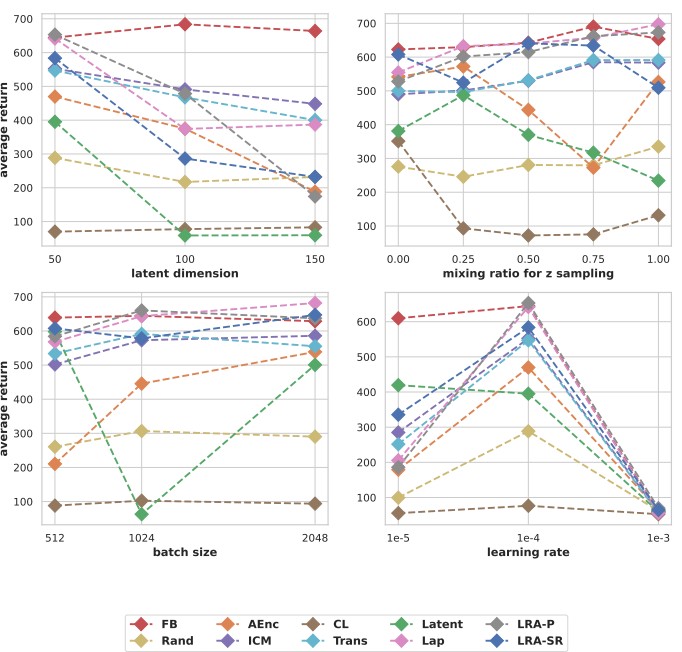

Figure 13: Return for each method trained in Walker domain on the RND replay buffer, for different choices of hyperparameters. Average over Walker tasks and 5 random seeds.

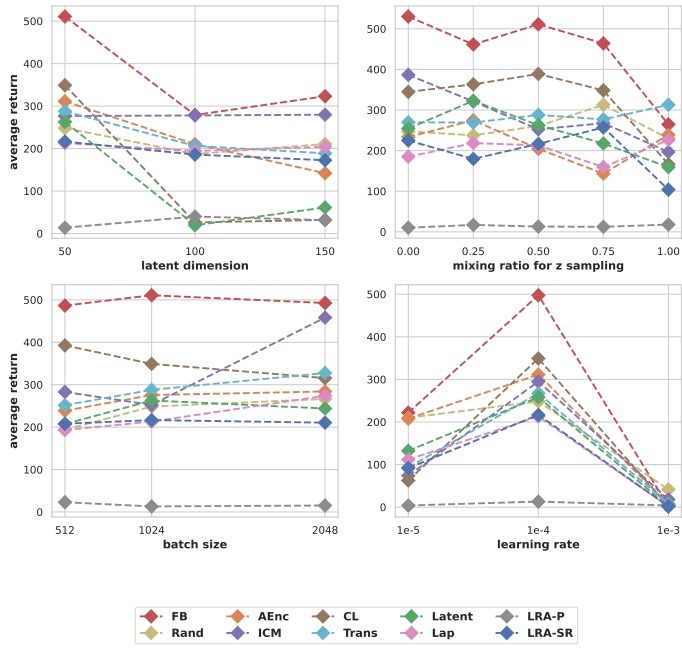

Figure 14: Return for each method trained in Cheetah domain, on the RND replay buffer, for different choices of hyperparameters. Average over Cheetah tasks and 5 random seeds.

## K    GOAL-ORIENTED BASELINES

Goal-oriented methods such as universal value functions (UVFs) (Schaul et al., 2015) learn a $Q$-function $Q(s, a, g)$ indexed by a goal description $g$. When taking for $g$ the set of all possible target states $s' \in S$, these methods can learn to reach arbitrary target states. For instance, Ma et al. (2020) use a mixture of universal SFs and UVFs to learn goal-conditioned agents and policies.

In principle, such goal-oriented methods are not designed to deal with dense rewards, which are linear combinations of goals $g$. Indeed, the optimal $Q$-function for a linear combination of goals is not the linear combination of the optimal $Q$-functions.

Nevertheless, we may still try to see how such linear combinations perform. The linear combination may be applied at the level of the $Q$-functions, or at the level of some goal descriptors, as follows.

Here, as an additional baseline for our experiments, we test a slight modification of the scheme from Ma et al. (2020), as suggested by one reviewer. We learn two embeddings $(\psi, w)$ using state-reaching tasks. The $Q$-function for reaching state $s'$ (goal $g = s'$) is modeled as

$$Q(s_t, a_t, s') = \psi(s_t, a_t, w(s'))^\top w(s'). \tag{44}$$

trained for reward $\mathbb{1}_{s_{t+1}=s'}$. A policy network $\pi(s, w(s'))$ outputs the action at $s$ for reaching goal $s'$. Thus, the training loss is

$$\mathcal{L}(\psi, w) := \mathbb{E}_{\substack{(s_t, a_t, s_{t+1}) \sim \rho \\ s' \sim \rho}} \left[ \psi(s_t, a_t, w(s'))^\top w(s') - \mathbb{1}_{\{s_{t+1}=s'\}} - \gamma \bar{\psi}(s_{t+1}, \pi(s_{t+1}, \bar{w}(s')), \bar{w}(s'))^\top \bar{w}(s') \right]^2 \tag{45}$$

Similarly to the other baselines, the policy network $\pi(s, w(s'))$ is trained by gradient ascent on the policy parameters to maximize

$$\mathbb{E}_{\substack{s \sim \rho \\ s' \sim \rho}} [Q(s, \pi(s, w(s')), s')] = \mathbb{E}_{\substack{s \sim \rho \\ s' \sim \rho}} \left[ \psi(s, \pi(s, w(s')), w(s'))^\top w(s') \right] \tag{46}$$

At test time, given a reward $r$, we proceed as in FB and estimate

$$z = \mathbb{E}_{s \sim \rho}[r(s)w(s)] \tag{47}$$

and use policy $\pi(s, z)$. This amounts to extending from goal-reaching tasks to dense tasks by linearity on $w$, even though in principle, this method should only optimize for single-goal rewards.

We use the same architectures for $(\psi, w)$ as for $(F, B)$, with double networks $\psi_1, \psi_2$ and policy smoothing as in (43).

The sparse reward $\mathbb{1}_{\{s_{t+1}=s'\}}$ could suffer from large variance in continuous state spaces. We mitigate this either by:

- biasing the sampling of $s'$ by setting $s'$ to $s_{t+1}$ half of the time.
- replacing the reward by a less sparse one, $\mathbb{1}_{\{\|s_{t+1}-s'\|_2 \leq \varepsilon\}}$.

Table 4 reports normalized scores for each domain, for the two variants just described, trained on the RND replay buffer, averaged over tasks and over 10 random seeds.

The first variant scores $84\%$ on Maze, $62\%$ on Quadruped tasks, $17\%$ on Walker tasks, and $2\%$ on Cheetah tasks. The second variant is worse overall.

So overall, this works well on Maze (as expected, since this is a goal-oriented problem), moderately well on Quadruped tasks (where most other methods work well), and poorly on the other environments. This is expected, as this method is not designed to handle dense combinations of goals.

**Relationship with FB.**    The use of universal SFs in Ma et al. (2020) is quite different from the use of universal SFs in Barreto et al. (2017) and Borsa et al. (2018). The latter is mathematically related to FB, as described at the end of Section 4. The former uses SFs as an intermediate tool for a goal-oriented model, and is more distantly related to FB (notably, it is designed to deal with single goals, not linear combinations of goals such as dense rewards).

A key difference between FB and goal-oriented methods is the following. Above, we use $\psi(s, a, w(g))^\top w(g)$ as a model of the optimal $Q$-function for the policy with goal $g$. This is reminiscent of $F(s, a, z)^\top B(g)$ with $z = B(g)$, the value of $z$ used to reach $g$ in FB.

| Reward | | | Domain | |
|---|---|---|---|---|
| | Maze | Walker | Cheetah | Quadruped |
| $\mathbb{1}_{\{s=s'\}}$ | 83.82 | 17.06 | 1.60 | 61.92 |
| $\mathbb{1}_{\{\|s-s'\|_2 \le \varepsilon\}}$ | 87.44 | 12.42 | 0.74 | 20.25 |

Table 4: Normalized score for each domain of two variants of USF from Ma et al. (2020) trained on RND replay buffer, averaged over tasks and 10 random seeds

However, even if we restrict FB to goal-reaching by using $z = B(g)$ only (a significant restriction), then FB learns $F(s, a, B(g))^\top B(g')$ to model the successor measure, i.e., the number of visits to each goal $g'$ for the policy with goal $g$, starting at $(s, a)$. Thus, FB learns an object indexed by $(s, a, g, g')$ for all pairs $(g, g')$.

Thus, FB learns more information (it models successor measures instead of $Q$-functions), and allows for recovering linear combinations of goals in a principled way. Meanwhile, even assuming perfect neural network optimization in goal-reaching methods, there is no reason the goal-oriented policies would be optimal for arbitrary linear combinations of goals, only for single goals.

## L  PSEUDOCODE OF TRAINING LOSSES

Here we provide PyTorch snippets for the key losses, notably the **FB** loss, **SF** loss as well as the various feature learning methods for **SF**.

```python
def compute_fb_loss(agent, obs, action, next_obs, z, discount):

  # compute target successor measure

  with torch.no_grad():
    mu = agent.policy_net(next_obs, z)
    next_action = TruncatedNormal(mu=mu, stddev=agent.cfg.stddev, clip=
    agent.cfg.stddev_clip)
    target_F1, target_F2 = agent.forward_target_net(next_obs, z,
    next_action)  # batch x z_dim
    target_B = agent.backward_target_net(next_obs)  # batch x z_dim
    target_M1, target_M2 = [torch.einsum('sd, td -> st', target_Fi,
    target_B) for target_Fi in [F1, F2]] # batch x batch
    target_M = torch.min(target_M1, target_M2)

  # compute the main FB loss

  F1, F2 = agent.forward_net(obs, z, action)
  B = agent.backward_net(next_obs)
  M1, M2 = [torch.einsum('sd, td -> st', Fi, B)  for Fi in [F1, F2]] #
    batch x batch
  I = torch.eye(*M1.size(), device=M1.device)
  off_diag = ~I.bool()
  fb_offdiag: tp.Any = 0.5 * sum((M - discount * target_M)[off_diag].pow
    (2).mean() for M in [M1, M2])
  fb_diag: tp.Any = -sum(M.diag().mean() for M in [M1, M2])
  fb_loss = fb_offdiag + fb_diag

  # compute the auxiliary loss

  next_Q1, nextQ2 = [torch.einsum('sd, sd -> s', target_Fi, z) for
    target_Fi in [target_F1, target_F2]]
  next_Q = torch.min(next_Q1, nextQ2)
  cov = torch.matmul(B.T, B) / B.shape[0]
  inv_cov = torch.linalg.pinv(cov)
  implicit_reward = (torch.matmul(B, inv_cov) * z).sum(dim=1) #
    batch_size
  target_Q = implicit_reward.detach() + discount * next_Q # batch_size

  Q1, Q2 = [torch.einsum('sd, sd -> s', Fi, z) for Fi in [F1, F2]]
  q_loss = F.mse_loss(Q1, target_Q) + F.mse_loss(Q2, target_Q)
  q_loss /= agent.cfg.z_dim
  fb_loss += q_loss

  # compute Orthonormality losss

  Cov = torch.matmul(B, B.T)
  orth_loss_diag = - 2 * Cov.diag().mean()
  orth_loss_offdiag = Cov[off_diag].pow(2).mean()
  orth_loss = orth_loss_offdiag + orth_loss_diag
  fb_loss += agent.cfg.ortho_coef * orth_loss

  return fb_loss
```

Listing 1: Pytorch code for **FB** training loss

```python
def compute_sf_loss(agent, obs, action, next_obs, z, discount):

  # compute target q-value
```

```python
4   with torch.no_grad():
5     mu = agent.policy_net(next_obs, z)
6     next_action = TruncatedNormal(mu=mu, stddev=agent.cfg.stddev, clip=
      agent.cfg.stddev_clip)
7     next_F1, next_F2 = agent.successor_target_net(next_obs, z,
      next_action)   # batch x z_dim
8     target_phi = agent.feature_net(next_goal).detach()   # batch x z_dim
9     next_Q1, next_Q2 = [torch.einsum('sd, sd -> s', next_Fi, z) for
      next_Fi in [next_F1, next_F2]]
10    next_Q = torch.min(next_Q1, next_Q2)
11    target_Q = torch.einsum('sd, sd -> s', target_phi, z) + discount *
      next_Q
12
13  F1, F2 = agent.successor_net(obs, z, action)
14  Q1, Q2 = [torch.einsum('sd, sd -> s', Fi, z) for Fi in [F1, F2]]
15  sf_loss = F.mse_loss(Q1, target_Q) + F.mse_loss(Q2, target_Q)
16
17  return sf_loss
```

Listing 2: Pytorch code for **SF** training loss

```python
1
2  def compute_phi_loss(agent, obs, next_obs):
3
4    phi = agent.feature_net(obs)
5    next_phi = agent.feature_net(next_obs)
6    loss = (phi - next_phi).pow(2).mean()
7
8    # compute Orthonormality losss
9
10   Cov = torch.matmul(phi, phi.T)
11   I = torch.eye(*Cov.size(), device=Cov.device)
12   off_diag = ~I.bool()
13   orth_loss_diag = - 2 * Cov.diag().mean()
14   orth_loss_offdiag = Cov[off_diag].pow(2).mean()
15   orth_loss = orth_loss_offdiag + orth_loss_diag
16
17   loss += orth_loss
18
19   return loss
```

Listing 3: Pytorch code for Laplacian Eigenfunctions **Lap** loss

```python
1  def compute_phi_loss(agent, obs, future_obs):
2
3    future_phi = agent.feature_net(future_obs)
4    mu = agent.mu_net(obs)
5    future_phi = F.normalize(future_phi, dim=1)
6    mu = F.normalize(mu, dim=1)
7    logits = torch.einsum('sd, td-> st', mu, future_phi)   # batch x batch
8    I = torch.eye(*logits.size(), device=logits.device)
9    off_diag = ~I.bool()
10   logits_off_diag = logits[off_diag].reshape(logits.shape[0], logits.
      shape[0] - 1)
11   loss = - logits.diag() + torch.logsumexp(logits_off_diag, dim=1)
12   loss = loss.mean()
13
14   return loss
```

Listing 4: Pytorch code for the contrastive **CL** loss

```python
1  def compute_phi_loss(agent, obs, action, next_obs):
2
3    phi = agent.feature_net(obs)
4    next_phi = agent.feature_net(next_obs)
```

```
5    predicted_action = agent.inverse_dynamic_net(torch.cat([phi, next_phi],
        dim=-1))
6    loss = (action - predicted_action).pow(2).mean()
7
8    return loss
```

Listing 5: Pytorch code of **ICM** loss

```
1 def compute_phi_loss(agent, obs, action, next_obs):
2
3    phi = agent.feature_net(obs)
4    predicted_next_obs = agent.forward_dynamic_net(torch.cat([phi, action],
        dim=-1))
5    loss = (predicted_next_obs - next_obs).pow(2).mean()
6
7    return loss
```

Listing 6: Pytorch code for **Trans** loss

```
1 def compute_phi_loss(agent, obs, action, next_obs):
2    phi = agent.feature_net(obs)
3    with torch.no_grad():
4      next_phi = agent.target_feature_net(next_obs)
5    predicted_next_obs = agent.forward_dynamic_net(torch.cat([phi, action],
        dim=-1))
6    loss = (predicted_next_obs - next_phi.detach()).pow(2).mean()
7
8    # update target network
9    for param, target_param in zip(agent.feature_net.parameters(), agent.
        target_feature_net.parameters()):
10     target_param.data.copy_(tau * param.data +
11              (1 - tau) * target_param.data)
12
13   return loss
```

Listing 7: Pytorch code for **Latent** loss

```
1 def compute_phi_loss(agent, obs):
2
3    phi = agent.feature_net(obs)
4    predicted_obs = agent.decoder(phi)
5    loss = (predicted_obs - obs).pow(2).mean()
6
7    return loss
8
```

Listing 8: Pytorch code for **AEnc** loss

```
1    def compute_phi_loss(agent, obs, action, next_obs):
2
3      phi = agent.feature_net(next_obs)
4      mu = agent.mu_net(torch.cat([obs, action], dim=1))
5      P = torch.einsum("sd, td -> st", mu, phi)
6      I = torch.eye(*P.size(), device=P.device)
7      off_diag = ~I.bool()
8      loss = - 2 * P.diag().mean() + P[off_diag].pow(2).mean()
9
10     # compute orthonormality loss
11     Cov = torch.matmul(phi, phi.T)
12     I = torch.eye(*Cov.size(), device=Cov.device)
13     off_diag = ~I.bool()
14     orth_loss_diag = - 2 * Cov.diag().mean()
15     orth_loss_offdiag = Cov[off_diag].pow(2).mean()
16     orth_loss = orth_loss_offdiag + orth_loss_diag
17     loss += orth_loss
18
```

```
19    return loss
20
```

Listing 9: Pytorch code for **LRA-P** loss

```
1  def compute_phi_loss(agent, obs, action, next_obs, discount):
2
3    phi = agent.feature_net(next_obs)
4    mu = agent.mu_net(obs)
5    SR = torch.einsum('sd, td -> st', mu, phi)
6    with torch.no_grad():
7      target_phi = agent.target_feature_net(next_obs)
8      target_mu = agent.target_mu_net(next_obs)
9      target_SR = torch.einsum("sd, td -> st", target_mu, target_phi)
10
11   I = torch.eye(*SR.size(), device=SR.device)
12   off_diag = ~I.bool()
13   loss = - 2 * SR.diag().mean()
14     + (SR - discount * target_SR.detach())[off_diag].pow(2).mean()
15
16   # compute orthonormality loss
17   Cov = torch.matmul(phi, phi.T)
18   orth_loss_diag = - 2 * Cov.diag().mean()
19   orth_loss_offdiag = Cov[off_diag].pow(2).mean()
20   orth_loss = orth_loss_offdiag + orth_loss_diag
21   loss += orth_loss
22
23   return loss
```

Listing 10: Pytorch code for **LRA-SR** loss

