# OpenReview forum: "Does Zero-Shot Reinforcement Learning Exist?"
_ICLR.cc/2023/Conference — ICLR 2023 notable top 25%_

### Official Review · Reviewer_uRSG · 2022-10-21

**Confidence:** 4
**Correctness:** 4
**Technical Novelty And Significance:** 2
**Empirical Novelty And Significance:** 3
**Recommendation:** 8

**Clarity, Quality, Novelty And Reproducibility:**

**[Clarity and Quality]**
- The paper is easy to follow and generally well-written.
- The second bullet point in Sec 1 is an incomplete sentence.

**[Novelty]**
- The technical novelty of the paper is limited as it focuses on empirical analysis of various SF/FB representations.
- The empirical results are novel to my knowledge.

**[Reproducibility]**
- Their code is not released.

**Strength And Weaknesses:**

**[Strengths]**
- To my knowledge, this is the first paper that systematically studies various choices for SF representations ($\phi$), which would be especially useful for practitioners who try to use SFs without relying on hand-crafted representations.
- The experiments are exhaustive and statistically sound (with 10 seeds).
- The claims in the paper are theoretically well-supported.

**[Weaknesses]**
- One thing that I do not fully agree with the authors is the distinction between SFs and FB. For me, the FB representation can be interpreted as a special case of USFs [1] with the tasks being all possible goal-reaching tasks (ignoring the subtlety in continuous domains). The authors distinguish FB from SFs in that SFs require predefined features and/or tasks (i.e., $\phi$), but I believe this is not accurate. For example, Ma et al. [1] use an end-to-end training scheme that learns a feature vector $\phi$ and a (learnable) task vector $w(g)$ by minimizing both the scalar loss (as in Eq (6)) and the vectorized loss (as in the equation above Eq (6)). If we assume that the source tasks consist of all the goal-reaching tasks to reach any states (and also assume a discrete state space), $w(g)$ becomes $B(s)$ and $\psi(s, a, z)$ becomes $F(s, a, z)$, and thus we exactly recover the FB objective. Hence, I believe the distinction between FB and SFs lies *not* in the way it automatically builds feature vectors ($\phi$) but rather in the loss function it uses (especially Eq (8)), which can handle continuous spaces as well, unlike USFs with all the goal-reaching tasks.
- Due to the reason above, I believe *learned* USFs (i.e., learnable $\phi(s)$ and $w(g)$ with the Q loss and/or vectorized loss [1]) trained with implicit goal-reaching tasks (from the dataset) should also be included as baselines. Also, some strong statements like "SFs must be provided with basic state features $\phi$" should be adjusted accordingly. I hope that the authors include detailed discussions and comparisons between FB and learned USFs.
- Since this paper focuses on the empirical study of SF representations, I believe the code should be made publicly available and hope that the authors release their code during the discussion period.

**[Additional questions]**
- What makes $\phi$ necessarily learn something in Eq (11) and (12)? It seems that $g$ or $f$ can capture everything with $\phi$ being the identity function.
- I didn't understand the following sentence in Sec 7: "Any low-rank model of $P_\pi$ will be poor; actually $P_\pi$ is better modeled as Id − low-rank, thus approximating the Laplacian.". Why does this amount to approximating the Laplacian? Also, it would be better to use parentheses around "low-rank" to avoid confusion.

[1] Chen Ma, Dylan R. Ashley, Junfeng Wen, and Yoshua Bengio. Universal Successor Features for Transfer Reinforcement Learning. 2020.


**Summary Of The Paper:**

The paper focuses on the empirical study of zero-shot RL, whose goal is to learn a set of policies and/or representations during training so that they can be adapted to solve unseen tasks in a zero-shot manner. Specifically, the authors focus on the successor features (SFs) framework and forward-backward (FB) representations, suggesting ten diverse choices for SF representations ($\phi$) as well as an improved mechanism to train FB representations. They test the SF/FB methods on 13 tasks from URLB with ExORL replay buffers and find that FB representations show consistently superior performances on the benchmark.

**Summary Of The Review:**

While there is a point (the distinction between FB and SFs) that I do not fully agree with the author's claim, I believe this paper is of high quality and will be helpful to the community. I thus recommend acceptance.

(Post-rebuttal update) Given the additional results and clarification, I raised my score to 8.

---

> ### Author Response · Authors · 2022-11-08
> **Answer to review by reviewer uRSG**
>
> Thank you for your comments, and in particular for your detailed questions about the relationship with USFs from [1].
>
> 1/ We do not believe FB can be recovered as a particular case of USFs from [1] with a different loss, even in the tabular case:
> - The objects have different types and argument sizes: identifying F and $\psi$, FB would be closer to $\psi(s,a,w)$ as in Borsa et al, not $\psi(s,a,g)$ as in [1].
> - After successful USF optimization from [1], we get optimal policies for all goals. After successful FB optimization we get optimal policies for all linear combinations of goals. If goals are target states, this is the difference between learning to reach arbitrary states and learning to optimize any reward, dense or sparse.
> - $\psi(s,a,g)^T w(g)$ only learns the optimal Q-value $Q(s,a,g)$ for reaching the goal or target state $g$, while $F(s,a,z)^T B(s')$ learns the successor measure $M^{\pi_z}(s,a,s')$ which contains much more information about $\pi_z$.
>
> 2/ It is not quite clear to us how you could use USFs from [1] as a baseline for general zero-shot RL. These USFs were introduced  to improve the generalization over goal space by training on a subset of goals and then finetuning on unseen goals at test time.  Therefore, this is only able to solve goal-reaching tasks, and is not shown to deal with general tasks as we present here.
>
> Notably, USFs from [1] need a goal $g$ to estimate $\psi(s,a,g)^T w(g)$ at test time. How would you set $g$ when facing a completely new, dense reward? Regressing the reward on the features provides a $w$ value, not a $g$ value. Could you tell us which algorithm you had in mind specifically for [1] as a baseline?
>
> 3/ Instead of USFs from [1], FB is closer to the USFs from Borsa et al, as discussed in Section 4. But even then, the theoretical correspondence between FB and SFs holds only under certain assumptions: a tabular or overparameterized model using fixed, nonlearnable B and phi, and after a change of variables. For general parameterizations (even linear), and for learnable B, the correspondence does not hold. Procedurally, FB and SFs are also quite different. All our SF baselines follow the same pattern: train the base features via some unsupervised criterion, then train successor features on top. FB has a different procedure, and putting it apart was clearest for the presentation of the paper.
>
> 4/ Sentence in Sec 7 wrt approximating the Laplacian: we will clarify the text. The Laplacian of a Markov chain is Id-P. Therefore, approximating P by Id-(low-rank) amounts to a low-rank approximation of the Laplacian.
>
> 5/ About phi in Eqs 11 and 12: we agree with your remark. Still, these are widely used this way, and we wanted standard methods for our baselines.
>
> As you say, in ICM there is indetermination between $\phi$ and $g$. Mathematically, $g$ could do all the work with $\phi=Id$. Still, in practice, we have a neural network in which the first layers are $\phi$ and the last layers are $g$: it is unlikely that the first layers will learn the identity and leave all the work to the last layers.
>
> 6/ We will definitely release the code, but we are facing hurdles between our institution's legal constraints and ICLR's anonymity constraint. In the meantime, we can provide pseudo-code for all the key functions and losses involved, notably for the various phi learning methods. Would that be satisfactory?  Are there further points needing clarification wrt implementation choices?

---

> > ### Comment · Reviewer_uRSG · 2022-11-10
> > **Response by Reviewer uRSG**
> >
> > 1, 2, 3/
> > Thank you for the response. However, I still believe FB is a special case of USFs [1] (in the tabular case) given the following conditions: (1) setting the goal space $\mathcal{G}$ to the entire state space $\mathcal{S}$ (i.e., goals can be all the possible states in the given MDP), (2) setting the reward $r$ to $\mathcal{I}(s=g)$, and (3) modeling $\psi(s, a, g)$ with $\psi(s, a, w(g))$ (this is actually different from the original USFs [1], but I believe it is a matter of architectural choice). With these, $\psi$ can be used to optimize any reward function, which can be viewed as a linear combination of all the state-reaching tasks, just as in FB representations. I acknowledge that the training procedure of FB is different from this version of USFs in the sense that it uses the loss specialized for modeling state occupancy (Eq (8)), which can also deal with a continuous state space, and I believe this is the main difference that distinguishes FB from SFs.
> >
> > Also, one possible way to use [1] for this offline zero-shot setting could be something like the following scheme:
> > (1) sample $(s, a, s')$ and $s+$ (which corresponds to a "goal") from the offline data,
> > (2) minimize $\lVert \psi(s, a, w(s+))^\top w(s+) - \mathcal{I}(s' = s+) - \gamma \psi(s', a'^*, w(s+))^\top w(s+) \rVert^2$ w.r.t. $\psi$ and $w$, and
> > (3) for a downstream task, use $\mathbb{E}[r(s) w(s)]$ (or regression if it also learns $\phi$) for the task vector in $\psi$, just as in SFs or FB.
> > I acknowledge that this procedure might not be the best way to utilize USF[1]-style training in this setting, but I believe it is still possible to have an SF baseline with learned $(\phi, \psi, w)$, and it would be very helpful to the community if the work additionally includes empirical comparisons between FB and some SF baseline with an end-to-end trained $(\phi, \psi, w)$.
> >
> > To clarify, my main concern is that, unlike the claim in the paper, SFs do not always require basic features $\phi$ but can also be end-to-end trained with learned $(\phi, \psi, w)$, as previously done in multiple works [1, 2, 3]. I hope that the paper discusses this aspect and provides a more accurate distinction between FB and SF. Specifically, I believe some statements such as "A ﬁrst difference between SFs and FB is that SFs must be provided with basic features $\phi$." should be at least qualified, e.g., by mentioning that it is also possible to use a learned $\phi$ with an end-to-end scheme and additionally discussing the feasibility of this end-to-end approach especially in this offline zero-shot setting, ideally with some empirical results.
> >
> > [1] Chen Ma, Dylan R. Ashley, Junfeng Wen, and Yoshua Bengio. Universal Successor Features for Transfer Reinforcement Learning. 2020. https://arxiv.org/abs/2001.04025
> >
> > [2] André Barreto, Shaobo Hou, Diana Borsa, David Silver, and Doina Precup. Fast Reinforcement Learning with Generalized Policy Updates. 2020. https://www.pnas.org/doi/10.1073/pnas.1907370117
> >
> > [3] Jaekyeom Kim, Seohong Park, Gunhee Kim. Constrained GPI for Zero-Shot Transfer in Reinforcement Learning. 2022. https://openreview.net/forum?id=sWNT5lT7l9G
> >
> > 4/ Thank you for the clarification.
> >
> > 5/ Thank you for the clarification and it sounds reasonable to me.
> >
> > 6/ I understand. It is fine if you release the code before the camera-ready deadline, and I believe there is no need for additional clarifications at the moment. That said, it would be nice if the final version of the paper contains pseudo-code for modified training procedures of FB and SFs with offline data.

---

> > > ### Author Response · Authors · 2022-11-10
> > > **Response to reviewer uRSG**
> > >
> > > Thank you for your quick and very detailed response!
> > >
> > > 1/ We do not believe that FB can be recovered by modifying USFs [1] as you suggest. The suggested update to [1] would lead to $\psi(s,a,w(g))^T w(g)$ as a model of the optimal Q-function for goal g. Meanwhile, even if we restrict FB to goal-reaching by using $z=B(g)$ only (a significant restriction), then FB learns $F(s,a,B(g))^T B(g')$ to model the number of visits to each g' for the policy with goal g, starting at (s,a). Thus FB learns an object indexed by (s,a,g,g') for all pairs (g,g'). This contains more information (it models successor measures instead of Q-functions), and allows for recovering linear combinations of goals in a principled way.
> > >
> > > Meanwhile, even assuming perfect neural network optimization in USFs [1], there is no theoretical reason the policies would be optimal for arbitrary linear combinations of goals, only for goal-oriented.
> > >
> > > 2/ Thank you for the detailed baseline suggestion. If we read this right, this amounts to UVFs as in Schaul et al 2015, trained for single goals, but extended to linear combinations w of goals at test time.
> > >
> > > We are adding this to the baselines and will report results in the coming days. Since we work with continuous spaces, we assume the sparse reward 1_{s'=s+} will be implemented as 1_{dist(s',s+)<=epsilon}. This would incur some high variance, and we would predict this to fail even with moderate dimension of s.
> > >
> > > 3/ We will clarify that indeed several works propose to learn phi automatically. We are a bit confused by your comments that we should include such comparisons, because this is precisely what 10 of our baselines do. Or are you referring specifically to (psi,phi,w)-learning versus (psi,phi)-learning baselines? In any case, we will include your baseline suggestion above in the final version.

---

> > > > ### Comment · Reviewer_uRSG · 2022-11-13
> > > > **Response by Reviewer uRSG**
> > > >
> > > > 1/ Thank you for the response. Now I see that FB can't be recovered as a special case of USFs [1]. It seems that on top of the aforementioned USF[1]-style scheme, FB has two additional features: (1) sampling both $z$ and $g$ for $\psi$ and $w$ respectively and (2) using a specialized reward function depending on both $z$ and $g$ so that it represents state occupancy measure. In light of this, I believe the "main" difference between FB and SFs lies in the fact that FB explicitly models state occupancy measure, and it consequently has more freedom to optimize this quantity (e.g., by using Eq (8) or Eq (9)). Since the current version of the paper appears to be claiming that the core difference between FB and SFs is whether they require basic feature $\phi$ or not, I hope that the paper provides a more accurate distinction between FB and SFs regarding this aspect.
> > > >
> > > > 2, 3/ My point is that it would be helpful if the work also shows empirical performances of *end-to-end* learned $(\phi, \psi, w)$ baselines (i.e., USF[1]-style), as opposed to the current ten baselines trained with *independent* objectives for $\phi$ (i.e., ICM). I believe such end-to-end baselines are the closest ones to FB, and thus comparing FB with them would be helpful to the community to better understand the difference between the two. I also agree that this sort of baseline may suffer from high variance (though it could be mitigated especially in this offline setting, as you can just set $s+$ to $s'$ with a fixed probability).

---

> > > > > ### Author Response · Authors · 2022-11-16
> > > > > **Response to Reviewer uRSG, new baseline**
> > > > >
> > > > > Thank you again for the responses!
> > > > >
> > > > > 1/ We got results for the baseline you suggested. We tested both the variant with $s+$ set to $s'$ part of the time to handle sparse rewards, and the variant with
> > > > > with $1_{dist(s,s')<\epsilon}$.
> > > > >
> > > > > The first variant scores 84% on Maze, 62% on Quadruped tasks, 17% on Walker tasks, and 2% on Cheetah tasks. The second variant is worse.
> > > > >
> > > > > So overall, this works well on Maze (as expected, since this is a goal-oriented problem), moderately well on Quadruped tasks (while most other methods work well), and poorly on the other environments. This is expected, as this method is not designed to handle dense rewards/combinations of goals.
> > > > >
> > > > > These results will be included in the paper.
> > > > >
> > > > > 2/ On the difference between SFs and FB being mostly the training of phi: this depends on the version of SFs we are talking about.
> > > > >
> > > > > SFs as in [1] (and UVFs as in Schaul et al 2015) deal with goal-oriented, and are quite different from FB (and they don't deal with linear combinations of goals).
> > > > >
> > > > > On the other hand, SFs as in Barreto et al 2017 or Borsa et al 2018 are mathematically closer to FB (though the correspondence is not trivial), and FB can be reinterpreted as a (complex) joint loss on psi and phi after a change of variables.
> > > > >
> > > > > We will clarify these variants of SFs in the text or appendix.

---

> > > > > > ### Comment · Reviewer_uRSG · 2022-11-16
> > > > > > **Response by Reviewer uRSG**
> > > > > >
> > > > > > Thank you for the additional experimental results. I believe these results and additional clarification would provide more insights to the community and thus I raised my score.

---

### Official Review · Reviewer_zaXC · 2022-10-23

**Confidence:** 3
**Correctness:** 2
**Technical Novelty And Significance:** 2
**Empirical Novelty And Significance:** 2
**Recommendation:** 3

**Clarity, Quality, Novelty And Reproducibility:**

**Clarity**

- The problem setting (called zero-shot RL in this paper) is unclear and inconsistent with what the actual experiment does. This paper defines zero-shot RL as follows: "The promise of zero-shot RL is to train **without rewards or tasks**, yet immediately perform well on any reward function given at test time, **with no extra training, planning, or finetuning**, and only a minimal amount of extra computation to process a task description (Section 2 gives a more precise definition)." I found this very confusing because of the following reasons.
  - Typically, zero-shot learning (or generalization) in RL has nothing to do with training in the absence of rewards or tasks. This setting is closer to **unsupervised** learning in RL but not zero-shot RL. While this paper indeed conducts unsupervised RL in the experiment, it is important to make it clear that this aspect is not about zero-shot RL.
  - "no extra training at test time" is consistent with typical zero-shot generalization in RL. However, the experiment in the paper violates this restriction by using 10,000 reward samples to train the reward-relevant component in SF and FB (or directly provide ground-truth $z_r$). Strictly speaking, this is not zero-shot generalization.
  - To my understanding, the actual problem setting considered in this paper can be described as "unsupervised representation learning with a offline RL dataset followed by fast adaptation to test tasks with a few samples of interactions". This setting is recently introduced by [Liu et al.] (cited in the paper). Although this is an interesting setting, I believe that this is not really about zero-shot learning.
- For the above reason, I think the current title is quite misleading.

**Quality**
- This paper provides a nice summary of successor features and forward-backward representation with useful insights.
- While this paper provided an extensive evaluation of existing representation learning methods for the specific problem, there is no further analysis showing why some methods are better or worse, other than several speculative hypotheses without empirical evidence in Section 7. It would be more convincing to show more in-depth analysis.

**Novelty**
- The empirical study conducted by this paper is new.
- There is no novel algorithm or method other than some interesting tricks to existing methods.

**Reproducibility**

While the paper provides hyperparameters in the appendix, there is not much detail for all of the unsupervised representation learning methods, which makes it hard to reproduce the result.

**Strength And Weaknesses:**

[Strength]

* Provides a nice survey of successor features, forward-backward representation, and unsupervised representation learning methods.

[Weakness]

* The definition of zero-shot RL in this paper is quite different from conventional zero-shot RL, which can be confusing.
* Although the evaluation of many methods is valuable, there is not much analysis or insight from the experiments.


**Summary Of The Paper:**

This paper conducts an empirical study of existing unsupervised pre-training methods for RL. Specifically, this paper focuses on successor features (SFs) and forward-backward representation (FB). The paper uses a variety of existing unsupervised representation learning methods as a way to provide state features for SFs. The empirical results on Unsupervised RL and ExORL benchmarks show that FB performs best.

**Summary Of The Review:**

Although this paper presents an interesting empirical study, the presentation of the paper needs to be significantly revised due to 1) the inconsistency between the problem definition and the experiment, and 2) the unconventional use of the term "zero-shot". In addition, the paper would benefit a lot from more in-depth analysis of existing feature learning methods.

---

> ### Author Response · Authors · 2022-11-10
> **Author answer to review by reviewer zaXC**
>
> Thank you for your review and the discussion about the choice of the word "zero-shot".
>
> 1/ Thank you for pointing out the need for further empirical analysis of why some methods fail. As suggested by another reviewer, we are adding tables of eigenvalues for the learned feature covariance matrix, as well as tSNE plots of the features learned. This provides more insight into some failures: clearly, some methods tend to exhibit feature collapse.
>
> 2/ About novelty: We agree that the FB method was introduced in Touati-Ollivier 2021, and that we only partly change its loss function (and clarify its links to SFs).
>
> But we consider that the promises of Touati-Ollivier 2021 were in need of more extensive experimental confirmation. We believe this is the first main contribution of this work. Second, we also believe this is the first extensive comparison of feature learning methods for successor features.
>
> 3/ About the word "zero-shot":
>
> Before choosing this word, we did a search for papers containing the expression "zero-shot reinforcement learning". Just a few papers showed up. They were all interesting, but also quite varied, and we got the impression that there was no established, widely accepted, formal definition of "zero-shot RL" in the literature.
>
> For us, the key feature of a zero-shot algorithm is that the models and neural networks are kept fixed at test time (no gradient descent, no update to the models). In this sense, we believe the experiments here qualify as "zero-shot".
>
> We do not agree that "the experiments[...] violates this restriction by using 10,000 reward samples to train the reward-relevant component in SF and FB": there is no training (no gradient step, no change to the model). The task has to be specified somehow: the 10,000 rewards encode the task by a direct weighted average, but no "training" occurs.
>
> Since we did not find a widely accepted formal definition, we chose to present in Section 2 a specific definition as used in this paper.
>
> We can clarify the definition earlier in the paper (abstract or intro instead of Section 2). Would this help?
>
> There are indeed two subcases:
> - unknown reward function, in which case a few reward samples are collected, but the models are not updated. This is few-shot in terms of rewards, and zero-shot in terms of updating or planning.
> - known reward function (including known goal, as in the Maze example): this is zero-shot in every sense.
>
> So these methods allow for zero-shot use cases.  We even have a small interactive demo (not part of the paper: we will only be able to release after anonymity is lifted) on the Walker environment, where the user can enter a reward function and have the agent instantly optimize it. We believe such use cases clearly qualify as "zero-shot".

---

### Official Review · Reviewer_UcNw · 2022-10-24

**Confidence:** 4
**Correctness:** 4
**Technical Novelty And Significance:** 3
**Empirical Novelty And Significance:** 4
**Recommendation:** 8

**Clarity, Quality, Novelty And Reproducibility:**

The paper is flawlessly written. All claims appear correct and well-defended. The work is novel and very significant.

Although details on the experimental setup are provided in the appendix, it is never the same as looking at the implementation choices and some hyperparameters are not mentioned. Releasing the code, the trained networks and the training logs would really make this paper reproducible and its results exploitable by others.

**Strength And Weaknesses:**

Disclaimer: given the very short timeline for reviewing all assigned papers, and to my great frustration, I didn't have time to check the appendix. Still, I browsed through it and it seems correct and (very) interesting. I did check the rest of the paper in detail and am confident in my evaluation.

### Strengths

This paper is very well written, easy to follow despite the technicality. One quality I find in the paper (which the authors do not claim) is that it provides a unified presentation of SR and FB. I believe this is incredibly useful for the community and is one important strength of such a paper. Because of this, I view Section 4 as a valuable contribution in itself, even though its contents may not be drastically novel (some of them actually are: the last paragraph of Section 4 on the link between FB and SF is very important, especially the argument explaining the avoidance of representation collapse).
Then, the formulations of Section 5 are useful and partially novel, especially those of Section 5.2 which are interesting reformulations of that of Touati & Ollivier (2021).
The empirical evaluation is thorough, well-conducted and well-discussed.
The main conclusion of the paper is that FB representations and Laplacian eigenfunctions for SR are more suited to zero-shot transfer. Limitations of the present evaluation are explicitly mentioned (although not really discussed). This conclusion, backed by convincing empirical evidence and relevant discussion is a nice contribution to the RL community.

### Weaknesses

This paper has a catchy name, but the name (although funny and I appreciate the pun) does not necessarily reflect the contribution: it is not really about existence of zero-shot RL but rather about the ability of neural networks to learn successor representations or measures. In the interest of readers in a few years, I would suggest keeping the catchyness but maybe focusing more on the ability to learn successor representations.

At the bottom of page 4, the authors present the fact that FB does not need state featurization as an advantage. This seems true from an abstract point of view, but I believe it deserves a bit more discussion. In very unbalanced datasets, or with very large variance transition models, or in the regime of very scarce data, fully relying on data to estimate occupancy measures boils down to discarding whatever expert prior knowledge might be available for state representations. In this case, the advantage of FB representations is less obvious and this begs for deeper evaluation (the discussion at the end of Section 7 somehow discards this question a bit too fast).

The AE features presented in Section 5.3 incur the risk of representation collapse in the sense that they might map to a set of linearly redundant features. From this perspective, I would have liked to have more insight as to the effective dimension of the learned feature space, for example as was done in the work of Lyle et al. (2022). There is a clear difference here since these features are not learned as solutions to a non-stationary regression problem, but still, the diversity of features could (should?) be made explicit.
In the same line of thought, only the latent transition model incorporates a term which discourages representation collapse (with a BYOL-like auxiliary loss). Maybe the other AE features would have benefited from such a term encouraging feature diversity (like regularizing features towards fixed or slowly moving random projections of the input state).
Lyle, C., Rowland, M., and Dabney, W. (2022). Understanding and preventing capacity loss in reinforcement learning. In 10th International Conference on Learning Representations.

Somehow connected to the previous remark, I would have appreciated a discussion on the latent spaces learned. Beyond the suggestion to evaluate the effective dimension of features (previous paragraph), I am curious as to what a tSNE of the replay buffer projected in the latent space would look like. Alternatively, the set of singular values from a PCA on the same set would be informative about the ability of the learned representation to shatter points. Since, in the discrete action case, the policy is eventually a linear classifier in the representation space, this latent space could (should) be better illustrated and discussed.

I slightly disagree with the idea that if the encoder-based losses learn nothing, it is because $(x,y)$ is already a good representation and they need to do nothing. They still have the freedom to learn bad (or collapsed) representations, which might be an alternative explanation. Also, in the same paragraph, it seems somehow counter-intuitive that SFs on (encoded) $(x,y)$ cannot learn the task: I am not convinced that planning requires "specific representations", I'd rather conclude that planning requires representations which preserve key information; this information is in the state $(x,y)$ but maybe not in its encoded version. The authors reiterate the claim that planning-specific features are necessary, in the paper's conclusion, and while I agree in the general case, I seem to fail to understand why the raw state is not a good planning specific feature since it "contains" the full state information necessary to build a policy. Is it because we need features which allow linear classification of actions? Maybe this deserves a clarification.

The paragraph describing how the scores of Figure 2 are computed is a bit unsettling. Since scores are normalized as a percentage of offline TD3, shouldn't the red dashed line stand at 100 on all figures? Is there a re-multiplication by the (average?) score of offline TD3?

I did not find a mention of the experimental code. Given the amount of experiments, releasing the code, the results and the training logs seems a crucial pre-requisite for fair dissemination of these results.

### Minor discussion elements

No need to address these in detail but I believe they could contain ideas for future work and I'm happy to share them and discuss them.

I am always a bit skeptical with the inclusion of $\gamma$ in the definition of an MDP. And even more skeptical when it comes to a reward-free MDP. This might sound a bit knitpicking but an MDP is primarily a stochastic process, whose definition does not include any notion of discount. Even the introduction of a reward model over the Markov chain conditioned by $s$ and $a$ does not require $\gamma$. It is the definition of the discounted return which involves $\gamma$ and this is quite independent of the MDP's dynamics or rewards: it is intrinsic to the criterion being maximized. Another criterion (average reward for instance) would not require $\gamma$ and the definition of the successor measure would not involve it either, while the reward-free MDP remains the same. Besides me being picky, this opens the door to alternative FB schemes for other criteria than the expected discounted sum of rewards.

There seems to be a striking connection between $z_r$ with successor features and the expression of policy values in LSTD / LSPI. This might sound a bit naive (and there are clear differences too) but I'm not sure this connection was made explicit before in prior work on SR.

**Summary Of The Paper:**

This paper studies zero-shot RL in the sense of "RL that does not require optimization of a policy when presented with a task (reward model)". It covers the literature on successor representations (SR) and forward-backward (FB) representation, proposes an unifying view and improved training losses. Then it empirically compares FB and many formulations of state features for SR, to assess which one provides (offline) representations which are suited for zero-shot deduction of (quasi-)optimal policies.

**Summary Of The Review:**

Overall, this paper is a strong contribution to learning successor representations and measures. The key finding is the general applicability of FB representations as successor measures and Laplacian eigenfunctions as successor features. Along the way, the paper provides a very appreciable unified view of learning representations for zero-shot RL, with insightful new formulations for FB representations.

---

> ### Author Response · Authors · 2022-11-14
> **Author response to review by reviewer UcNw**
>
> Thank you very much for your review and appreciation of our work, and in particular for the in-depth discussion of feature collapse. We will start with this point.
>
> 1/ About our statement that "planning requires specific representations": we apologize, this should read "planning *via successor features* requires specific features", and we will update the text and conclusion.
>
> The raw state (x,y) is not a good representation for planning via successor features, because successor features based on phi=(x,y) can only optimize rewards of the form a.x+b.y.
>
> On the other hand, (x,y) contains all the information necessary for auto-encoders or transition models: such methods have no incentive to learn anything beyond (x,y) (maybe up to some bijective change of variables). They could even, as you say, learn worse representations and erase information. But a tSNE plot confirms that AEnc and Trans preserve (x,y) info perfectly.
>
> We apologize for the unclear statement, as we believe we actually agree on the matter itself.
>
> 2/ Latent space and feature collapse via effective rank and tSNE:
>
> We analyzed the effective rank of the features as in the reference you pointed out (singular values above a certain threshold). We will include a table in the Appendix. For most methods and all environments except Maze, more than 90% (often 100%) of singular values are above 1e-4, so the effective rank is close to full.
>
> The Maze environment is a clear exception: on Maze, for the Rand, AEnc, Trans, Latent and ICM methods, only one third of the singular values are above 1e-4. The other methods keep 100% of the singular values above 1e-4. This is perfectly aligned with the performance of each method on Maze.
>
> In the Maze environment, auto-encoder or transition models can perfectly optimize their loss just by keeping the original two features (x,y), and they have no incentive to learn other features, so the effective rank could have been 2. Forcing against eigenvalue collapse would not necessarily improve things, we believe, because this could just be done by keeping (x,y) and then blowing up some irrelevant features.
>
> So rank reduction does happen for some methods. But this just reflects the fact that two features (x,y) already convey the necessary information about states and dynamics, but are not sufficient to solve the problem via successor features.
>
> We also have tSNE visualizations of learned features for the Maze environment (the most easily interpreted by visual inspection).
>
> AEnc, Trans, FB, LRA-P and CL all recover a nice picture of the maze. For the other methods the picture is less clear (notably, Rand gets a circle, which could just be an instance of concentration of measure theorems).
>
> Whether tSNE preserves the shape of the maze does not appear to be correlated to performance: AEnc and Trans learn nice features but perform poorly, while the tSNE of Lap is not visually interpretable but performance is good.
>
> These singular value and tSNE figures will be included in the updated version of the appendix.
>
> 3/ About using state featurization with FB: this point is discussed in Touati-Ollivier 2021 (paragraph "Incorporating prior information on rewards"). If prior knowledge is available, then B can use user-provided features phi as input instead of the full state. This makes FB learning easier, and gives access to any reward that can be expressed as functions of phi (linear or not, contrary to SF). So, indeed, FB could exploit any prior state featurization. Space permitting, we will add a remark to that effect.
>
> 4/ Scores in Fig 1 and Fig 2: Fig 1 reports normalized scores, and Fig 2 reports absolute scores. We apologize for the confusion and will clarify the text.
>
> We wanted normalized scores in Fig 1 to express zero-shot RL scores as a fraction of the topline (maximal supervised scores on the same trainset), and also to standardize before aggregating over tasks and environments. We decided to provide raw, non-normalized information for Fig 2 because this is important information to keep.
>
> 5/ A link between LSTD and successor measures can be found in the thesis of Leonard Blier, "Some Principled Methods for Deep Reinforcement Learning".
>
> 6/ We will definitely release the code, but we are facing hurdles between our institution's legal constraints and ICLR's anonymity constraint. In the meantime, we can provide pseudo-code for all the key functions and losses involved, notably for the various phi learning methods. Would that be satisfactory? Are there further points needing clarification wrt implementation choices?
>
> 7/ Title and existence of zero-shot RL: we do believe our work is primarily about whether proposed zero-shot RL methods are effective or not. Notably, we consider that the zero-shot promises of Touati-Ollivier 2021 were in need of more extensive experimental confirmation. We believe one of the main contributions of the present work is to confirm that proposed zero-shot methods can be effective.

---

### Official Review · Reviewer_NRQ8 · 2022-10-24

**Confidence:** 4
**Correctness:** 4
**Technical Novelty And Significance:** 3
**Empirical Novelty And Significance:** 3
**Recommendation:** 10

**Clarity, Quality, Novelty And Reproducibility:**

Quality
- Paper shows a systematic comparison of prior unsupervised RL methods, provides methodology and empirical comparison and discussions.
- The findings are mostly interesting and could be helpful for future research in building zero-shot RL models from large unlabelled datasets.

Clarify
- Mostly very clear.
- The key concepts are introduced clearly and the main claims are well supported.


**Strength And Weaknesses:**

Strength
- This work systematically tested forward-backward representations and many new models of successor features on zero-shot tasks. The evaluation on the ExoRL dataset and unsupervised RL benchmark is very extensive and across multiple different qualities of datasets. It shows that FB representation performs best and consistently across the board in a zero-shot manner.
- The writing is well-written and well-organized, the authors introduce prior work and compare them both in both methodology and experimental aspects. The discussion of limitations is thoughtful and contains interesting points.
- This work systematically evaluates prior zero-shot RL methods in the same setting, and then proposes a novel and effective method that consistently outperforms prior works, reaching 85% of supervised RL performance with a good replay buffer, in a zero-shot manner.
- The results are significant and will facilitate future research on unsupervised zero-shot RL.

Weaknesses
- The biggest weakness seems to be that it assumes there is offline diverse data. One one hand, this is a limitation since unlike NLP there is a large unlabeled dataset naturally available, in RL, collecting diverse dataset is still not yet solved. On the other hand, this work focuses on evaluating learning zero-shot RL from pre collected unlabeled dataset, which is an important research question and it’s reasonable to have such assumptions in order to make the evaluation easier.


**Summary Of The Paper:**

Paper shows a systematic study of different strategies for approximate zero-shot RL including successor features (SFs) and forward-backward (FB) representations. It introduces the concepts clearly and discusses the advantages and limitations of each method in the two categories. Paper introduces new SF models and compares them in zero-shot RL tasks from the Unsupervised RL benchmark.

**Summary Of The Review:**

The authors study zero-shot RL that pretrained on large unlabeled datasets for different downstream tasks, they conduct a systematical extensive evaluation of various unsupervised RL methods, and reveal algebraic links between SFs, FB, contrastive learning, and spectral methods. Seems to me that the findings in this paper are important for future research.

---

> ### Author Response · Authors · 2022-11-14
> **Author response to review by reviewer NRQ8**
>
> Thank you very much for your review and for your positive comments!
>
> We agree that all of the methods tested rely on training data diversity. These methods deal with learning good representations for generic planning, but not with exploration. This is why we used three different off-the-shelf exploration methods to test robustness to this aspect. Finding the best exploration methods for FB/SFs, or even using FB and SFs to enhance exploration, are among the next topics we will investigate.

---

### Author Response · Authors · 2022-11-18
**Updated appendix with feature rank analysis, tSNE, additional baseline, and code snippets**

We have just updated the appendix to include the additional information and analysis suggested by the reviewers:

- feature rank analysis to test feature collapse
- tSNE plots of the learned representations
- additional goal-oriented baselines as suggested by reviewer uRSG
- code snippets for the loss functions of the various methods.

We will also update the main text accordingly.

---

### Decision · Program_Chairs · 2023-01-20

**Decision:**

Accept: notable-top-25%

**Justification For Why Not Higher Score:**

While the results in the paper are great, the experiments are only conducted on a specific class of methods (Mujoco continuous control in deterministic environments). The paper would be more impactful if it could also demonstrate similar results in other types of environments (e.g. game-like environments). Additionally, it assumes access to samples of the reward function, and a good pre-training distribution, which somewhat limit the generality.

**Justification For Why Not Lower Score:**

The results in the paper are very impressive (85% of what supervised offline TD3 achieves, *without* requiring rewards during pretraining), and it will no doubt be an impactful paper for the RL community.

**Metareview: Summary, Strengths And Weaknesses:**

This paper defines the problem of "zero-shot RL", in which an agent should be able to solve any reward function in an environment with no further learning or planning, given (1) unsupervised (reward-free) experience in that same environment and (2) access to the true reward function at inference time. The paper examines two approaches for zero-shot RL, one based on successor features and one based on forward-backward representations. The paper empirically evaluates these approaches on the unsupervised RL benchmark suite, demonstrating that while SF approaches perform inconsistently, the FB approach performs well on all tasks and approaches performance of an agent trained from scratch with rewards.

The reviewers were generally positive about the paper, citing the systematic, extensive evaluation of different SF representation learning methods plus FB and the strong, significant results. The reviewers initially had requests for clarifications which were addressed in the rebuttal both via explanation and the addition of some further experiments and analysis. I found the results very impressive and believe this will be an impactful paper if it is presented at ICLR; I therefore recommend acceptance.

**Note From Pc:**

if the above contains the word "oral" or "spotlight" please see: "oral" presentation means -> notable-top-5% and "spotlight" means -> notable-top-25%. As stated in our emails, we are disassociating presentation type from AC recommendations

**Summary Of Ac-Reviewer Meeting:**

N/A